# β-cell intrinsic dynamics rather than gap junction structure dictates subpopulations in the islet functional network

Jennifer K Briggs[1], Anne Gresch[1,2], Isabella Marinelli[3], JaeAnn M Dwulet[1], David J Albers[1,4], Vira Kravets[1], Richard KP Benninger[1,2]*

[1]Department of Bioengineering, University of Colorado Anschutz Medical Campus, Aurora, United States; [2]Barbara Davis Center for Childhood Diabetes, University of Colorado Anschutz Medical Campus, Aurora, United States; [3]Centre for Systems Modelling and Quantitative Biomedicine, University of Birmingham, Birmingham, United Kingdom; [4]Department of Biomedical Informatics, University of Colorado Anschutz Medical Campus, Aurora, United States

**Abstract** Diabetes is caused by the inability of electrically coupled, functionally heterogeneous β-cells within the pancreatic islet to provide adequate insulin secretion. Functional networks have been used to represent synchronized oscillatory [Ca$^{2+}$] dynamics and to study β-cell subpopulations, which play an important role in driving islet function. The mechanism by which highly synchronized β-cell subpopulations drive islet function is unclear. We used experimental and computational techniques to investigate the relationship between functional networks, structural (gap junction) networks, and intrinsic β-cell dynamics in slow and fast oscillating islets. Highly synchronized subpopulations in the functional network were differentiated by intrinsic dynamics, including metabolic activity and K$_{ATP}$ channel conductance, more than structural coupling. Consistent with this, intrinsic dynamics were more predictive of high synchronization in the islet functional network as compared to high levels of structural coupling. Finally, dysfunction of gap junctions, which can occur in diabetes, caused decreases in the efficiency and clustering of the functional network. These results indicate that intrinsic dynamics rather than structure drive connections in the functional network and highly synchronized subpopulations, but gap junctions are still essential for overall network efficiency. These findings deepen our interpretation of functional networks and the formation of functional subpopulations in dynamic tissues such as the islet.

*For correspondence: richard.benninger@cuanschutz.edu

Competing interest: The authors declare that no competing interests exist.

## Editor's evaluation

The paper uses both computational and laboratory approaches to test the hypothesis that connectivity in β cells within the islet is due to metabolic rather than gap junctional coupling efficacy. This will be an important advance for understanding the role of heterogeneous β cell populations in driving synchronized oscillations by islets and by extension the oscillatory insulin secretion observed in vivo. There will be implications of the work for understanding the mechanisms of type 2 diabetics and β cell function in general.

## Introduction

Diabetes mellitus is a global epidemic afflicting >500M adults world-wide (*Sun et al., 2022*) and is associated with dysfunction or death of insulin-producing β-cells within pancreatic islets. β-cells are electrically coupled through connexin36 (Cx36) gap junctions (*Benninger et al., 2008*; *Serre-Beinier*

*et al., 2000*; *Ravier et al., 2005*), and are functionally heterogeneous. Electrical coupling and heterogeneity are both central to regulating the homeostasis of glucose and other nutrients (*Benninger and Kravets, 2022*; *Pørksen et al., 2002*; *Matveyenko et al., 2012*; *Bratusch-Marrain et al., 1986*). Gap junction electrical coupling is essential for synchronizing β-cell electrical dynamics and allowing insulin to be released in robust coordinated pulses (*Benninger et al., 2008*; *Benninger and Kravets, 2022*; *Weitz et al., 2021*; *Head et al., 2012*). Functional heterogeneity allows improved islet robustness (*Dwulet et al., 2021*) and control of islet pulsatile dynamics (*Benninger and Kravets, 2022*; *Dwulet et al., 2021*; *Dwulet et al., 2019*). In diabetes, there can be a lack of intra-islet β-cell communication (*Matveyenko et al., 2012*; *Satin et al., 2015*; *Farnsworth and Benninger, 2014*) and an insufficient number of adequately functioning β-cells (*Matveyenko et al., 2012*; *Bratusch-Marrain et al., 1986*; *Weitz et al., 2021*; *Head et al., 2012*). Therefore, the role of islet communication, β-cell heterogeneity, and collective synchronization is fundamental to understanding islet dysfunction in diabetes.

Network analysis is a tool used to quantify relationships between interacting parts of a system (*Koutrouli et al., 2020*; *Strogatz, 2001*; *Newman, 2003*; *Watts and Strogatz, 1998*; *Lynn and Bassett, 2019*), represented by its entities (nodes) and their interactions (edges). The islet can be studied using *structural* or *functional* networks. In a *structural network,* edges represent physical conduits for communication (*Lynn and Bassett, 2019*; *Bullmore and Sporns, 2009*; *Gansterer et al.,*

**Table 1.** Essential network statistics and types.

Left: Five network statistics used to quantify the network in this paper. Representative networks show an example of the statistic in red and the rest of the network in blue. Right: Five network types referred to in this paper. Regular, small world, and random networks are all made with the same number of nodes and edges, but different configurations of edges. Scale free network shows three "hub" nodes where node size is proportional to degree.

**Network essentials**

| Network statistics | | | Network types | | |
|---|---|---|---|---|---|
| Degree | Number of edges for a given node |  | Regular | A network with ordered connections: high clustering coefficient and long average path length |  |
| Degree distribution | The distribution of the network's degrees | | Small world | A regular network with a few rewired edges: high clustering coefficient and short average path length |  |
| Clustering coefficient | Likelihood of how often neighbors of a node share connections with each other $$\frac{3\ (\text{\# of triangles})}{(\text{\# of connected triples})}$$ |  | Random | A network with random connections has low clustering coefficient and short average path length |  |
| Shortest path | Shortest distance between any two nodes |  | Scale-free | A network whose degree distribution follows a power law, such that there are a few very highly connected nodes called hubs |  |
| Efficiency | Inverse of the shortest path | | Weighted | A network whose edges are weighted by some edge property |  |

2019; *Rings et al., 2022*), which include gap junctions in the islet. However, islet networks are often studied using a *functional network* representation, where edges connect β-cell pairs that have highly correlated [Ca²⁺] dynamics (*Lynn and Bassett, 2019*; *Chialvo, 2010*; *Stožer et al., 2013*; *Johnston et al., 2016*). β-Cell functional networks have been suggested to follow a scale-free distribution (*Stožer et al., 2013*) with high clustering (*Gosak et al., 2018*; *Markovič et al., 2015*) and 'small-world-like qualities' (see *Table 1*). In small world networks, there exists a subset of highly synchronized nodes (referred to as 'β-cell hubs' in the islet; *Bullmore and Sporns, 2009*; *Johnston et al., 2016*), which have stronger than average influence over network dynamics. Optogenetic-mediated hyper-polarization of these β-cell hubs has been demonstrated to perturb overall islet coordination, and these hubs are disrupted under diabetogenic conditions (*Johnston et al., 2016*). Therefore, in theory, preferential removal of these hubs will stop the network from functioning. However, it has been questioned whether this small world topology is possible in the islet, as individual cells (or small subpopulations of cells) do not have enough electrical influence to control the entire islet (*Dwulet et al., 2021*; *Satin et al., 2020*; *Rutter et al., 2020*).

Oscillations of [Ca²⁺] in β-cells are driven by the activity of glucokinase, closure of ATP-sensitive K⁺ (K$_{ATP}$) channels, and subsequent bursts of action potentials. Therefore, synchronized [Ca²⁺] dynamics, and the corresponding functional network, may be influenced by both gap junction communication and factors influencing intrinsic cell dynamics such as glucokinase and K$_{ATP}$ activity. Because debate exists over the topology of the islet network and whether β-cell subpopulations can dictate islet dynamics, an important question is how the functional network and β-cell hubs are driven by structural communication compared to intrinsic cell dynamics. This question is important in our understanding of islet dynamics in diabetes because both the gap junctions (e.g. the structural network) (*St. Clair et al., 2020*; *Corezola do Amaral et al., 2020*; *Farnsworth et al., 2022*; *Hodson et al., 2013*; *Carvalho et al., 2012*; *Farnsworth et al., 2016*) and synchronization (e.g. the functional network) (*Pørksen et al., 2002*; *Satin et al., 2015*; *Corezola do Amaral et al., 2020*; *Hodson et al., 2013*; *Haefliger et al., 2013*) can be impaired in diabetic conditions.

To investigate the relationship between functional networks, structural networks, and the intrinsic dynamics of the β-cells, we framed our study around three questions. Our first question asks, *are highly correlated subpopulations (β-cell hubs) within the functional network differentiated with respect to the structural network or by their intrinsic properties?* While sometimes assumed, it is unclear whether β-cell hubs show significantly greater structural connections or whether they are differentiated by their intrinsic dynamics. Our second question asks: *what does the islet functional network indicate about its underlying structure or the intrinsic dynamics of individual cells?* That is, are edges within the functional network representing strong structural connections or pathways between β-cell nodes? Finally, as gap junctions may become dysfunctional in diabetic conditions, our third question asks: *how do changes to the structural network influence the islet functional network?* By answering these questions, we provide insight into the relationship between structure, function, and intrinsic cell dynamics in the pancreatic islet.

## Results

### Simulations of fast Ca²⁺ oscillations reveal that β-cell hubs have a more pronounced difference in metabolic activity than gap junction coupling

To investigate the relationships between functional and structural networks, we first analyzed a simulated β-cell network. We used a well-validated multi-cellular coupled ODE model, which describes the electrophysiological properties of the β-cells (*Dwulet et al., 2019*; *Notary et al., 2016*; *Hraha et al., 2014*; *Westacott et al., 2017*) and results in fast calcium oscillations (<2 min). This model is based on the Cha-Noma model (*Cha et al., 2011*) and contains detailed kinetics of the main ion channels in the islet that affect membrane potential, including the voltage-gated Ca²⁺ channels (Ca$_V$), K$_{ATP}$ channels, or store-operated currents. We include heterogeneity in parameters underlying cellular excitability, such as the rate of glucokinase ($k_{glyc}$) and maximum K$_{ATP}$ conductance ($g_{KATP}$). We also included heterogeneity in the gap junction electrical coupling ($g_{coup}$) (*Figure 1a*). The model was simulated at 11 mM glucose.

Our first question asks whether subpopulations that emerge from the β-cell functional network are driven by the structural network or by intrinsic properties of the cells. We extracted the islet functional

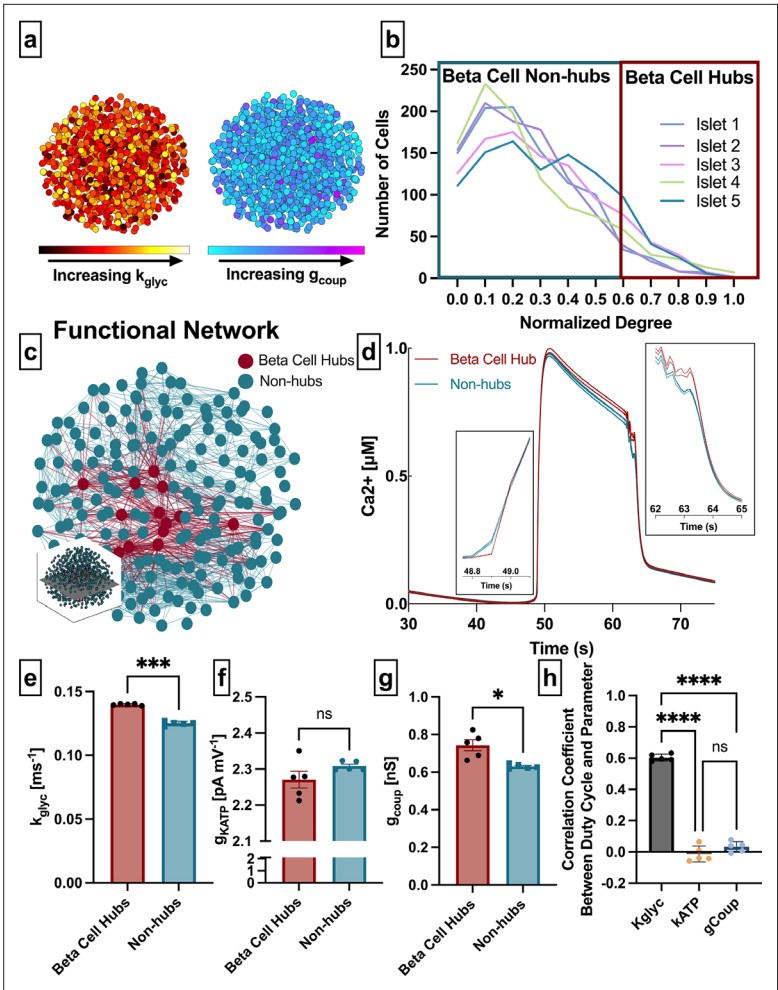

**Figure 1.** Analysis of parameters underlying functionally connected cells from the Cha-Noma model, representing fast oscillations. (**a**) Schematic of 1000 β-cell computational model, with cells false-colored by heterogeneity in rate of glucokinase, $k_{glyc}$ (left) and gap junction conductance, $g_{coup}$ (right) parameter values. (**b**) Distribution of functional connections (edges), determined from functional network analysis. Colors show five simulated islets. Hub cells (red outline) are any cell with >60% of maximum number of links. (**c**) Two-dimensional slice of the simulated islet, with lines (edges) representing functional connections between synchronized cells. Hub cells indicated in red. Slice is taken from middle of islet (see inset). (**d**) Representative [Ca²⁺] time course of a hub (red) and non-hub (blue). (**e**) Average rate of glucose metabolism ($k_{glyc}$) values compared between hubs and non-hubs in a Cha-Noma simulated islet. Each data point represents the averaged values for a single islet. Effect size was 2.85. (**f**) As in e for maximum conductance of ATP-sensitive potassium channel ($g_{KATP}$). Effect size was 0.31. (**g**) As in e for gap junction conductance ($g_{coup}$). Effect size was 0.82. (**h**) Pearson correlation coefficient between duty cycle and $k_{glyc}$, $g_{KATP}$, and $g_{coup}$ for all cells. Significance in e–g was determined by paired Student's t-test. *p ≤ 0.05 and ***p ≤ 0.001. In e–g each data point corresponds to the average value over a single simulated islet. Error bars represent s.e.m.

The online version of this article includes the following figure supplement(s) for figure 1:

**Figure supplement 1.** Functional network sensitivity to threshold in Cha-Noma model.

**Figure supplement 2.** Functional network dependence with alternative hub cell definition in Cha-Noma model.

network by assigning a node to each cell and an edge between any cell pair for which the intracellular free Ca²⁺ ([Ca²⁺]) time course showed a correlation coefficient greater than a threshold ($R_{th}$). The degree was defined as the number of edges a given node possesses (**Table 1**). In order to study similar subpopulations as in previous studies, we set the threshold $R_{th}$ to match previous studies which show a normalized degree distribution that includes many cells with a low degree and few cells with a high degree (**Stožer et al., 2013**; **Johnston et al., 2016**) (see Methods) (**Figure 1b** and **Table 1**). We then identified β-cell hubs as cells in which the normalized functional network degree was greater than

60%, in accordance with prior studies that identified β-cell hubs (*Johnston et al., 2016*; *Figure 1b–d*). β-Cell hubs made up 5–12% of the coupled Cha-Noma model.

We examined the relationship between functional network-derived β-cell hubs and parameters impacting their intrinsic dynamics. Glucokinase conversion of glucose to glucose 6-phosphate is the rate-limiting step in glycolysis, therefore the rate of glucokinase activity ($k_{glyc}$) is a proxy for the metabolic activity in the cell. The maximal open-channel $K_{ATP}$ conductance ($g_{KATP}$) is equivalent to the number of $K_{ATP}$ channels, which also influences β-cell dynamics. β-Cell hubs had a significantly higher $k_{glyc}$ than non-hubs (mean± SD 0.14±0.0007 ms$^{-1}$ vs 0.12±0.0017 ms$^{-1}$, p<0.001) (*Figure 1e*) but did not differ in $g_{KATP}$ (2.27±0.05 nS vs. 2.31±0.01 nS, p=0.18) (*Figure 1f*). These results indicate that in the coupled Cha-Noma model, β-cell hubs are determined by their metabolic activity more than solely the number of $K_{ATP}$ channels. Interestingly, β-cell hubs had only a slightly higher $g_{coup}$ than non-hubs (0.74±0.07 pS vs. 0.63±0.01 pS, p=0.032) (*Figure 1g*). To investigate the influence of these parameters on Ca$^{2+}$ dynamics in this model, we correlated the parameter values with [Ca$^{2+}$] oscillation duty cycle for each simulated islet. In the coupled Cha-Noma model, $k_{glyc}$ was more strongly correlated with duty cycle than $g_{KATP}$ (p<0.0001) and $g_{coup}$ (p<0.0001) (*Figure 1h*). Therefore, β-cell hubs were most strongly differentiated by $k_{glyc}$, which in turn was strongly correlated with the [Ca$^{2+}$] oscillation duty cycle.

To ensure consistency in our findings, we repeated the analysis with several thresholds ($R_{th}$) (*Figure 1—figure supplement 1a*). For all thresholds analyzed, the difference between hubs and non-hubs was greater for $k_{glyc}$ than for $g_{coup}$ (*Figure 1—figure supplement 1b and c*), with $k_{glyc}$ being significantly higher in hubs for all five thresholds and $g_{coup}$ being significantly higher in hubs for only two thresholds. We also used an alternative method to identify β-cell hubs, as the 10% of cells with the highest network degree. With this alternative definition, β-cell hubs still had a significantly higher $k_{glyc}$, a slightly higher $g_{coup}$ and similar $g_{KATP}$ as compared to non-hubs (*Figure 1—figure supplement 2*).

These results indicate that both rate of glucokinase activity and gap junction coupling correlate with high cellular synchronization, where the rate of glucokinase activity has a much stronger correlation.

## Simulations of slow Ca$^{2+}$ oscillations reveal that $K_{ATP}$ channel conductance and gap junction coupling are distinct features of β-cell hubs

In addition to fast oscillations modeled in *Figure 1*, β-cells exhibit slow Ca$^{2+}$ oscillations with periods greater than 2 min (*Liu et al., 1998*). We next used the integrated oscillator model (IOM) (*Marinelli et al., 2021*; *Marinelli et al., 2022*) that describes details of the metabolic oscillations that underlie slow Ca$^{2+}$ oscillations to test whether β-cell hubs emerging from a model of slow oscillations were differentiated by their gap junction coupling or intrinsic dynamics. Unlike the Cha-Noma model (*Figure 1*), the IOM has not previously been coupled using more than two cells, which is necessary for studying heterogeneity. We formed a coupled IOM, again with heterogeneity in parameters describing the rate of glucokinase activity ($k_{glyc}$) (referred to as $v_{GK}$ in previous manuscripts; *Marinelli et al., 2021*; *Marinelli et al., 2022*), maximum $K_{ATP}$ conductance ($g_{KATP}$), and structural gap junction electrical coupling ($g_{coup}$) and simulated at 11 mM glucose. Due to limitations in computational power, we limited the islets to 260 cells (*Figure 2a*). We again chose a threshold to obtain a 'scale-free-like' distribution (*Figure 2b and c*). β-Cell hubs made up 3–7% of the coupled IOM.

β-Cell hubs did not differ in $k_{glyc}$ compared to non-hubs (mean ± SD 3.67±0.47 ms$^{-1}$ vs 3.36±0.07 ms$^{-1}$, p=0.19) (*Figure 2d*). However, β-cell hubs had significantly lower $g_{KATP}$ (20.1±0.3 nS vs. 20.4±0.1 nS, p=0.04) (*Figure 2e*) and significantly higher $g_{coup}$ (9.5±1.2 pS vs. 6.0±0.1 pS, p=0.0028) (*Figure 2f*). These findings were less stable than the Cha-Noma model when the threshold $R_{th}$ was altered (*Figure 2—figure supplement 1*). When identifying β-cell hubs as the 10% of cells with the highest network degree, hubs still had significantly higher $g_{coup}$ but similar $g_{KATP}$ and $k_{glyc}$ compared to non-hubs (*Figure 2—figure supplement 2*).

To test why hubs in the coupled IOM were determined by $g_{KATP}$ more than $k_{glyc}$ (as in the Cha-Noma model), we again correlated duty cycle with parameter value. As opposed to the coupled Cha-Noma model (*Figure 1h*), in the coupled IOM, duty cycle was strongly negatively correlated with $g_{KATP}$ and weakly positively correlated with $k_{glyc}$ ($g_{KATP}$ vs. $k_{glyc}$: p<0.0001 and $g_{coup}$ vs $k_{glyc}$: p=0.0002) (*Figure 2g*). Therefore, independent of oscillation type and computational model, the parameter that was most correlated with duty cycle was also most influential in determining hubs in the functional network.

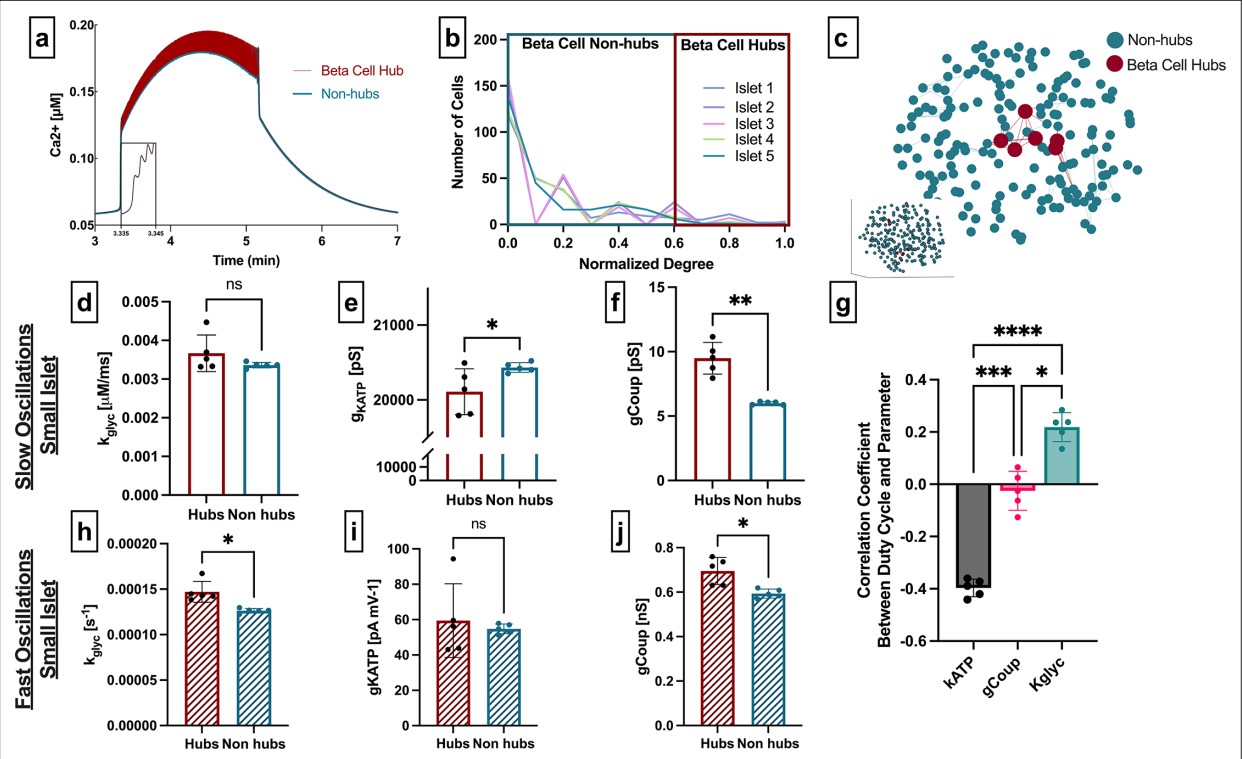

**Figure 2.** Analysis of parameters underlying functionally connected cells from the integrated oscillator model, representing slow oscillations. (**a**) Two-dimensional slice of the simulated islet from slow simulation, with lines (edges) representing functional connections between synchronized cells. Hub cells indicated in red. Slice is taken from middle of islet (see inset). (**b**) Distribution of functional connections (edges), determined from functional network analysis. Colors show five simulated islets. Hub cells (red outline) are any cell with >60% of maximum number of links. (**c**) Representative $[Ca^{2+}]$ time course of a hub (red) and non-hub (blue) from slow islet simulation. (**d**) Average rate of glucokinase ($k_{glyc}$) values compared between hubs and non-hubs in a slow simulated islet, retrospectively analyzed. Each data point represents the averaged values for a whole islet. Effect size was 0.65. (**e**) As in d for maximum conductance of ATP-sensitive potassium channel ($g_{KATP}$). Effect size was 1.27. (**f**) As in d for gap junction conductance ($g_{coup}$). Effect size was 2.94. (**g**) Pearson correlation coefficient between duty cycle and $k_{glyc}$, $g_{KATP}$, and $g_{coup}$ for all cells. (**h**) As in d but for fast model with 260 cells. (**i**) As in e but for fast model with 260 cells. (**j**) As in f but for fast model with 260 cells. Significance in d–j was determined by paired Student's t-test. *p≤0.05 and **p≤0.01. In d–j each data point corresponds to the average value over a single simulated islet. Error bars represent s.e.m.

The online version of this article includes the following figure supplement(s) for figure 2:

**Figure supplement 1.** Functional network sensitivity to threshold in integrated oscillator model (IOM).

**Figure supplement 2.** Functional network dependence with alternative hub cell definition in integrated oscillator model.

To enable direct comparisons between the coupled Cha-Noma and coupled IOM, we re-simulated the coupled Cha-Noma model with similar numbers of cells as the IOM. In the smaller coupled Cha-Noma model, the influence of $k_{glyc}$ on β-cell hubs became more similar to the influence of $g_{coup}$ (***Figure 2h and j***). There was still no significant difference in $g_{KATP}$ for β-cell hubs and non-hubs (***Figure 2i***). These results suggest that as the islet becomes smaller, the influence of gap junction coupling in determining β-cell hubs grows to be comparable to the intrinsic dynamics ($k_{glyc}$, $g_{KATP}$) that drive $Ca^{2+}$ oscillation duty cycle.

## Elevated metabolic activity, but not elevated gap junction permeability, is observed experimentally in highly synchronized cells

We next applied experimental approaches to investigate the β-cell functional network and the properties of highly synchronized cells. We performed time-lapse imaging of $[Ca^{2+}]$ within islets isolated from Mip-Cre[ER];Rosa-LSL-GCamP6s mice that express GCamP6s in β-cells (β-GCamP6s mice, see Methods). We identified β-cells that showed synchronous slow oscillations in $[Ca^{2+}]$ (***Figure 3a and b***). We extracted the functional network from $[Ca^{2+}]$ oscillations in the second phase using a threshold of $R_{th} = 0.9$ for all islets analyzed (***Figure 3c***), where this threshold was chosen to reflect a scale-free-like

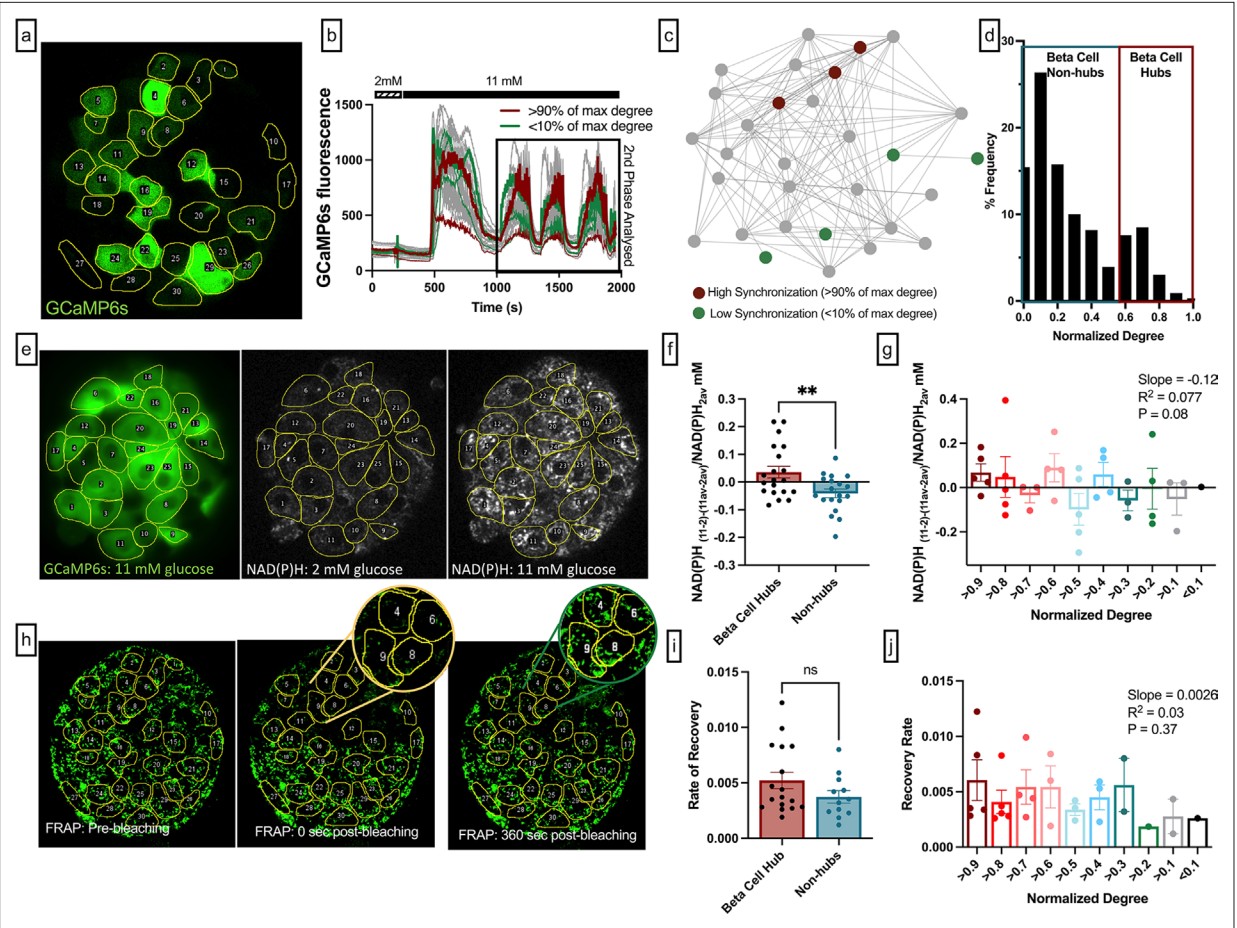

**Figure 3.** Experimental comparison between the functional network from slow β-cell oscillations, NAD(P)H activity, and coupling conductance.
(**a**) Mouse pancreatic islet expressing fluorescent calcium indicator GCaMP6s in β-cells. Glucose level 11 mM. (**b**) GCamp6s time traces recorded at 2 and 11 mM glucose. Red curves represent dynamics of the most coordinated cells. These cells had highest number of edges, i.e., normalized degree >0.9. Green curves represent dynamics of the least coordinated cells, i.e., normalized degree <0.1, the rest of the cells are shown in gray. Only second phase (shown in black box) was used for functional network analysis. (**c**) Ca²⁺-based functional network of the islet shown in (**e**). Red dots represent most coordinated cells, and green dots – least coordinated cells which had at least 1 edge. (**d**) Degree distribution of all 11 analyzed islets. Threshold of $R_{th}$ = 0.9 was used to filter the Pearson-based coordination matrix and obtain the functional network. (**e**) *Left:* Mouse pancreatic islet expressing fluorescent calcium indicator GCaMP6s. Glucose level 11 mM. *Middle:* NAD(P)H autofluorescence of the islet and the same cell layer, recorded at 2 mM glucose. *Right:* NAD(P)H autofluorescence recorded at 11 mM glucose. (**f**) Change of the NAD(P)H levels in each cell in response to glucose elevation, with respect to the islet average change. Metabolic activity here is compared between β-cell hubs and non-hubs. (**g**) Change in NAD(P)H levels in response to glucose elevation, with respect to the islet average change: here comparison is made for the most coordinated cells (normalized degree >0.9) with the less coordinated cells (normalized degree >0.8, >0.7, >0.6, …,<0.2, <0.1). Linear regression resulted in a trending relationship between normalized degree and NAD(P)H with slope of –0.126 (p=0.079). (f, g) represent n=5 islets with a total of 131 cells. Each dot represents the average NAD(P)H level for cells with respective degrees in an islet. (**h**) Rhodamine-123 fluorescence of the islet and the same cell layer as in (**a**), recorded immediately before photobleaching of the top half of each islet (left, red), immediately after photobleaching (middle, yellow) and 360 s after the photobleaching (right, green). (i) Fluorescence recovery after photobleaching (FRAP) recovery rate (s⁻¹) between β-cell hubs and non-hubs. (**j**) FRAP recovery rate (s⁻¹) in each of the photobleached cells: here the comparison is made for the most coordinated cells (normalized degree >90%) with the less coordinated cells (normalized degree >80, >70, >60, …<20,<10%). Linear regression resulted in a non-significant relationship between normalized degree and recovery rate (p=0.29). (**i, j**) shows results from 9 islets. Data points represent the average recovery rate for cells with respective degrees in an islet. **p≤0.01. In all panels, error bars represent s.e.m.

The online version of this article includes the following figure supplement(s) for figure 3:

**Figure supplement 1.** Experimental results for fast and mixed oscillations reveal no relationship between gap junction conductance and functional degree.

**Figure supplement 2.** Experimental results with alternative hub cell definition.

distribution for most islets (*Figure 3d*). Our results showed no correlation between network degree and mean GCamP6 fluorescence intensity (p>0.8). β-Cell hubs (>60% of max degree) made up 3–40% of islets.

To test whether metabolic activity was correlated with the functional network degree, we measured two-photon excited NAD(P)H autofluorescence in conjunction with time-lapse imaging of [Ca²⁺] dynamics at 2 mM and 11 mM glucose (*Figure 3e*). The metabolic response of individual β-cells in the islet was calculated as the difference between NAD(P)H autofluorescence at high glucose compared to low glucose ($NAD(P)H_{11-2}$), with the islet average subtracted to account for inter-islet variability ($NAD(P)H_{11-2} - NAD(P)H_{11av-2av}$). This metric was 0 if the NAD(P)H response of a cell was the same as the islet average, >0 if a cell showed an elevated metabolic response, and <0 if a cell showed a reduced metabolic response. β-Cell hubs (>60% of max degree) had a significantly higher NAD(P)H autofluorescence than in non-hubs (<60% of max degree) (p=0.0061) (*Figure 3f*). Similarly, the NAD(P)H response trended toward a relationship with cell's normalized degree (p=0.079) (*Figure 3g*). When identifying β-cell hubs as the 10% of cells with the highest network degree, hubs (top 10% by degree) also had a significantly higher NAD(P)H autofluorescence than in non-hubs (bottom 90% by degree) (*Figure 3—figure supplement 2a*). This indicates that highly synchronized cells are more metabolically active than less synchronized cells, in agreement with findings from simulated islets in the coupled Cha-Noma model (*Figure 1*).

The relationship between gap junction coupling and the functional network is not known. Fluorescence recovery after photobleaching (FRAP) measurements of dye transfer kinetics can quantify gap junction permeability (*Farnsworth et al., 2014*). We performed FRAP measurements in conjunction with time-lapse imaging of [Ca²⁺] dynamics, to map cellular gap junction connections in the same cell

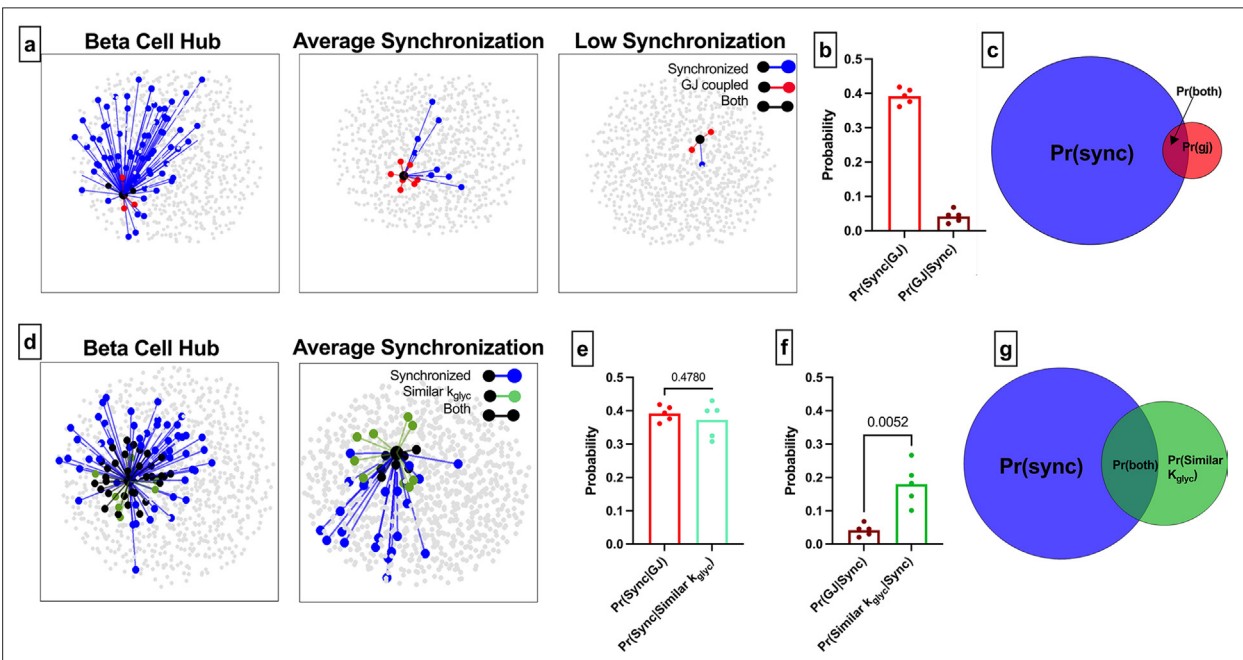

**Figure 4.** Comparison of functional connections, gap junctions, and glucose metabolism in the simulated islet. (**a**) Functional connections and structural connections for a representative highly connected β-cell hub, average cell, and low connected cell. Blue cells and edges indicate cells determined as 'synchronized' via functional network analysis. Red cells are GJ coupled. Black cells are both GJ coupled and synchronized. (**b**) Probability that a cell is synchronized given it is gap junction coupled, or vice versa, that a cell is gap junction coupled given it is synchronized. (**c**) Venn diagram showing overlap between the synchronized cells and gap junction coupled cells within the simulated islet. (**d**) Functional connections and metabolic connections for a β-cell hub and a cell with averaged synchronization. Blue cells and edges indicate cells determined as 'synchronized' via functional network analysis. Green cells and edges indicate cells that have similar $k_{glyc}$ and are within 15% islet from each other. Black cells and edges indicate both synchronized and similar $k_{glyc}$. (**e**) Probability that a cell pair is synchronized given it shares a gap junction has similar $k_{glyc}$. (**f**) Probability that the cell pair is gap junction coupled has similar $k_{glyc}$ and within 15% of islet distance (*Pr*=0.0052) , respectively, given the pair is synchronized. (**g**) Venn diagram showing overlap between the synchronized cells and metabolically similar cells. Shaded area in c and g is proportional to indicated probability. Significance in e and f was determined by a repeated measures one-way ANOVA with Tukey's multiple comparisons **p≤0.01. In b, e, f each data point corresponds to the average value over a single simulated islet.

layer as the functional network (*Figure 3h*). There was no difference in the rate of recovery for β-cell hubs and non-hubs (p=0.15) (*Figure 3i*). Similarly, the rate of recovery, which assesses gap junction permeability, was not statistically related to the normalized degree (p=0.37) (*Figure 3j*). The lack of relationship between rate of recovery and normalized degree was maintained over different functional network synchronization thresholds (*Figure 3—figure supplement 1a and b*). We also repeated these analyses in islets that showed fast oscillations or mixed oscillations. Irrespective of oscillation type, there was no difference between the rate of recovery for β-cell hubs and non-hubs, nor was there an association between functional network degree and the rate of recovery (*Figure 3—figure supplement 1c–h*). When identifying β-cell hubs as the 10% of cells with the highest network degree, there was also no difference between the rate of recovery for β-cell hubs and non-hubs (*Figure 3—figure supplement 2b*). These results indicate that cell synchronization is not correlated with gap junction permeability, as measured by FRAP. *Therefore, the functional network is not strongly related to the structural network.*

## Synchronization between a cell pair more likely indicates shared intrinsic properties than elevated gap junction coupling in simulated islets

We next used the coupled Cha-Noma simulated islet to investigate the second question: what does the islet functional network indicate about its underlying structure or intrinsic properties, on an individual cell basis? Irrespective of the functional network degree of a cell, it was rare for a cell pair to be connected in both the functional and structural networks (*Figure 4a*). The probability that two cells were synchronized in the functional network, given that they shared a gap junction in the structural network, was Pr(*Sync│GJ*)=0.39 (*Figure 4b*). Therefore, the presence of a structural edge does not imply a functional edge. Importantly, when the threshold $R_{th}$ was decreased, the probability of synchronization given gap junction increased to almost 1.0 (*Figure 1—figure supplement 1d and e*). This indicates, as expected, that gap junctions are required for synchronizing cell pairs. However, additional similarities are required for a cell pair to be synchronized enough to surpass a threshold large enough to cause the functional network to appear 'scale-free-like' that is demonstrated in prior studies (*Stožer et al., 2013*; *Johnston et al., 2016*).

The alternative probability of a structural edge given a functional edge was Pr*(GJ │Sync)*=0.04 (*Figure 4b*), indicating that very few functionally coupled cells are connected by gap junctions. These findings indicate that structurally connected cells and functionally connected cells are two distinct groups with little overlap (*Figure 4c*). We analyzed the sensitivity of these measures to the synchronization threshold $R_{th}$. Tautologically, as the threshold was increased, the number of functionally connected cells decreased (*Figure 1—figure supplement 1d*), causing Pr*(GJ│Sync)* to increase (*Figure 1—figure supplement 1f*). However, the 'overlap' between the functional network and structural network did not increase (*Figure 1—figure supplement 1g*). Our findings that the functional and structural networks are two distinct groups with little overlap is consistent across thresholds.

These data corroborate with initial findings showing the functional network is not strongly correlated with the structural network. To investigate whether $k_{glyc}$ was associated with edges in the functional network, we created a new network where edges were drawn between cells with similar $k_{glyc}$ rates (see Methods) (*Figure 4d*). Parameters were chosen such that there was no difference between Pr*(Sync│Similar $k_{glyc}$)* and Pr*(Sync│GJ)* (*Figure 4e*), allowing for a direct comparison between probabilities. The alternative probability Pr*(Similar $k_{glyc}$│Sync)* was significantly higher than Pr*(GJ│Sync)* (*Figure 4f*). If two cells have synchronized $Ca^{2+}$ oscillations, they are more likely to contain shared rate of glucokinase activity than to be gap junction coupled. This is further indicated by the increased overlap between the $[Ca^{2+}]$ synchronization-derived functional network and the intrinsic rate of glucokinase activity-derived network (*Figure 4g*). These results further indicate that intrinsic cell dynamics is a greater driving factor than gap junction connections for cells to show high $[Ca^{2+}]$ synchronization and influence the functional network.

## Elevated metabolic activity is a greater driver of long-range functional connections in simulated islets

In accordance with the islet cytoarchitecture, gap junction connections only exist between highly proximal cells. However, there is no distance constraint on $[Ca^{2+}]$ oscillation synchronization (*Figure 5a*).

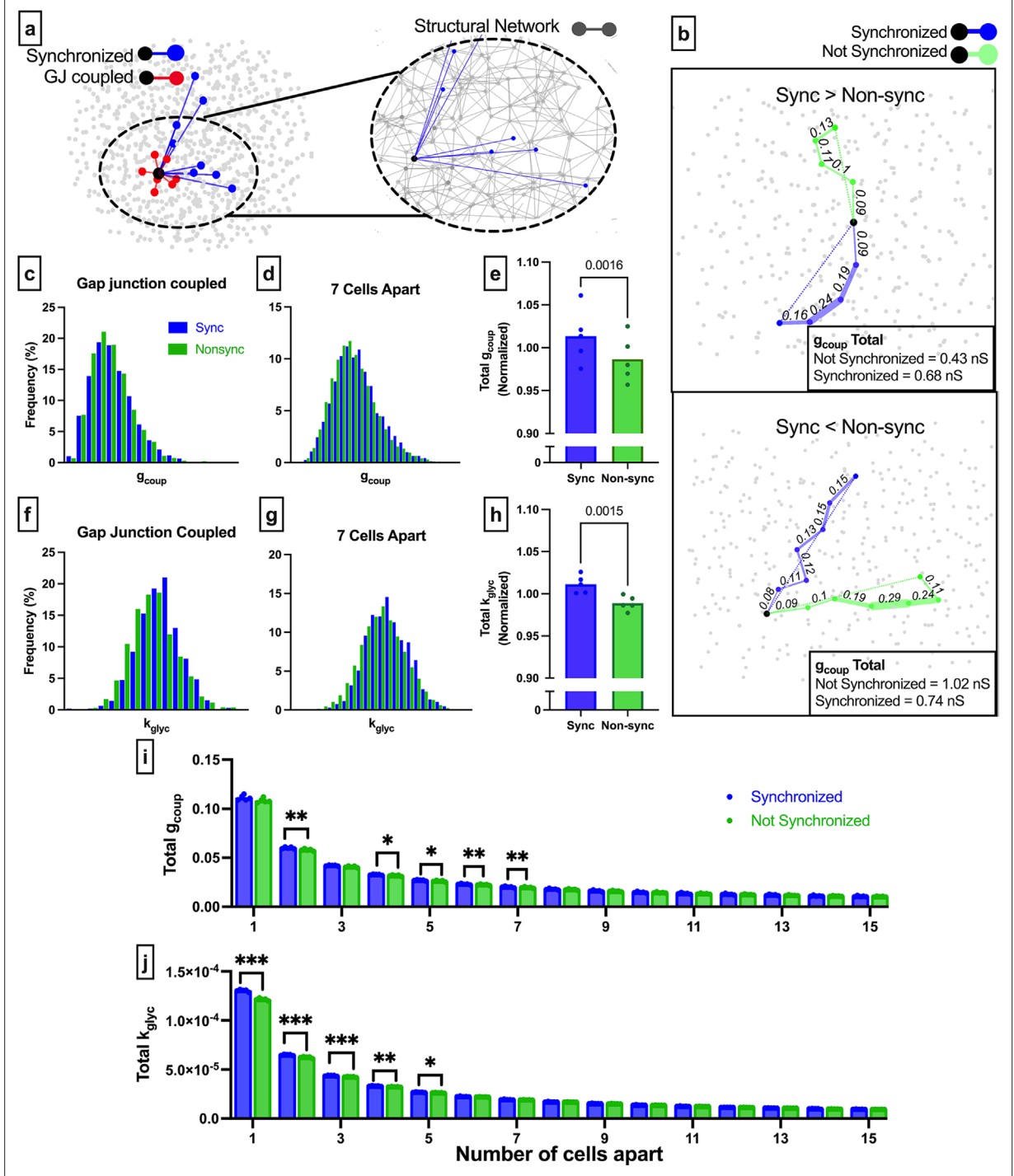

**Figure 5.** Comparison of long-range functional synchronization, gap junction network, and glucose metabolism in the simulated islet. (**a**) Functional connections and gap junction connections for a representative cell (black). Synchronized cells in blue, gap junction coupled cells in red. Inset shows the entire structural network (gray) with functional connections for the same cell shown in blue. (**b**) Two representative cells (black) and the shortest path to a synchronized cell (blue) and a non-synchronized cell (green). Path is weighted (shown by edge thickness) by $g_{coup}$. For the cell on the top panel, the synchronized path has a higher cumulative gap junction conductivity (0.68 nS) than the non-synchronized path (0.43 nS). For the cell on the bottom panel, the synchronized path has a lower cumulative gap junction conductance (0.74 nS) than the non-synchronized path (1.02 nS). (**c**) Probability distribution of total $g_{coup}$ for synchronized cells (blue) and non-synchronized cells (green) that are directly connected by gap junctions. (**d**) As in c for cells pairs that are 7 cells apart. (**e**) Comparison of the total $g_{coup}$, normalized by distance for synchronized cells (blue) and non-synchronized cells (green). Each dot indicates the average resistance for a single simulated islet. (**f**) As in c but with connections weighted by rate of glucokinase activity ($k_{glyc}$). (**g**) As in d but with connections weighted by $k_{glyc}$. (**h**) As in e but with connections weighted by $k_{glyc}$. (**i**) Total $g_{coup}$ for synchronized and non-synchronized

*Figure 5 continued on next page*

*Figure 5 continued*

cells organized by cell distance. (**j**) As in i but for the network weighted by $k_{glyc}$. Significance in e, h was assessed by a two-tailed paired t-test, with p-value indicated. Significance in i and j was assessed by paired t-tests with Bonferroni correction. Reported p-values are adjusted. *p≤0.05, **p≤0.01, ***p≤0.001. In (**e, h, i**, j) each data point corresponds to the average value for synchronized/non-synchronized cell pairs (**e, h**) or respective number of cells apart over a single simulated islet.

The online version of this article includes the following figure supplement(s) for figure 5:

**Figure supplement 1.** Paths that maximized gap junction conductivity.

**Figure supplement 2.** Paths that maximized metabolic rate.

To determine whether this spatial constraint was responsible for the low correspondence between the functional and structural networks, we asked whether a series of highly conductive gap junction connections could predict long-range functional connections. We weighted the structural network by $g_{coup}$ and calculated the path which maximized conductance (see Methods) (***Figure 5b***). Not all synchronized cell pairs had chains of larger total $g_{coup}$ than non-synchronized cell pairs (***Figure 5b***). The probability distributions of total $g_{coup}$ for synchronized cells highly overlapped with those for non-synchronized cells, irrespective of distance (***Figure 5c and d***, and ***Figure 5—figure supplement 1a***). However, the total conductance normalized by separation distance was significantly less for non-synchronized cells than for synchronized cells (***Figure 5e***). Thus, on average, long-range functional connections traverse cells that are connected by higher gap junction conductance.

To examine how the intrinsic rate of glucokinase activity influences long-range functional connections, we repeated this procedure but weighted graph edges by $k_{glyc}$ rather than $g_{coup}$. Again, the probability distributions of the total $k_{glyc}$ for synchronized cells overlapped with those for non-synchronized cells, irrespective of distance (***Figure 5f and g*** and ***Figure 5—figure supplement 2***). The total rate of glucokinase activity, normalized by the separation distance, was significantly less for non-synchronized cells than for synchronized cells (***Figure 5h***). Thus, on average, long-range functional connections also traverse cells with higher metabolic activity.

To test whether there was a spatial relationship for rate of glucokinase activity or gap junction conductance-controlled synchronization, we analyzed the total conductance or total rate of glucose metabolism for different separation distances and between synchronized or non-synchronized cell pairs. Total $g_{coup}$ was significantly higher for synchronized cells 4–7 cells apart (***Figure 5i***). Total $k_{glyc}$ was significantly higher for synchronized cells 1–5 cells apart, with the significance larger than that of total $g_{coup}$ (***Figure 5j***). Thus, cell pairs with similar $k_{glyc}$ can strongly influence functional connections up to 5 cells apart, while high $g_{coup}$ is necessary for influencing functional connections over longer distances (5–7 cells apart).

## Decreasing gap junction conductance in the structural network decreases the average network degree in the functional network constructed based on the experimental and simulated data

Our third question asks how changes to the structural network influence the islet functional network. Reducing Cx36 gap junction coupling via genetic or pharmacological means reduces overall islet $Ca^{2+}$ oscillation synchronization (***Benninger et al., 2008***; ***Farnsworth and Benninger, 2014***). We performed functional network analysis on islets from wild-type (WT) mice (Cx36[+/+]), heterozygous Cx36 knockout mice with ~50% reduced gap junction coupling (Cx36[+/-]), and homozygous Cx36 knockout mice with no gap junction coupling (Cx36[-/-]) (***Benninger et al., 2008***; ***Figure 6a–b***). We used a single synchronization threshold $R_{th}$ across all genotypes so the average degree distribution of WT islets roughly followed a scale-free-like distribution (***Stožer et al., 2013***) and had an average of between 5 and 15 connections (see Methods) (***Figure 6—figure supplement 1***). Cx36[+/-] islets demonstrated decreased average network degree compared to WT Cx36[+/+] islets (***Figure 6c***), and homozygous Cx36[-/-] demonstrated further decreased degree compared to both WT Cx36[+/+] islets and Cx36[+/-] islets (***Figure 6c***). Therefore, the functional network becomes highly sparse as edges in the structural network decrease.

We next quantified how the removal of edges in the structural network influences the functional network topology (***Table 1***). Network topology statistics can provide insight into how the network functions, and how it changes with specific perturbations. *Clustering coefficient ($C_{avg}$)* quantifies the proportion of nodes connected to a given node that are also connected with each other (***Table 1***).

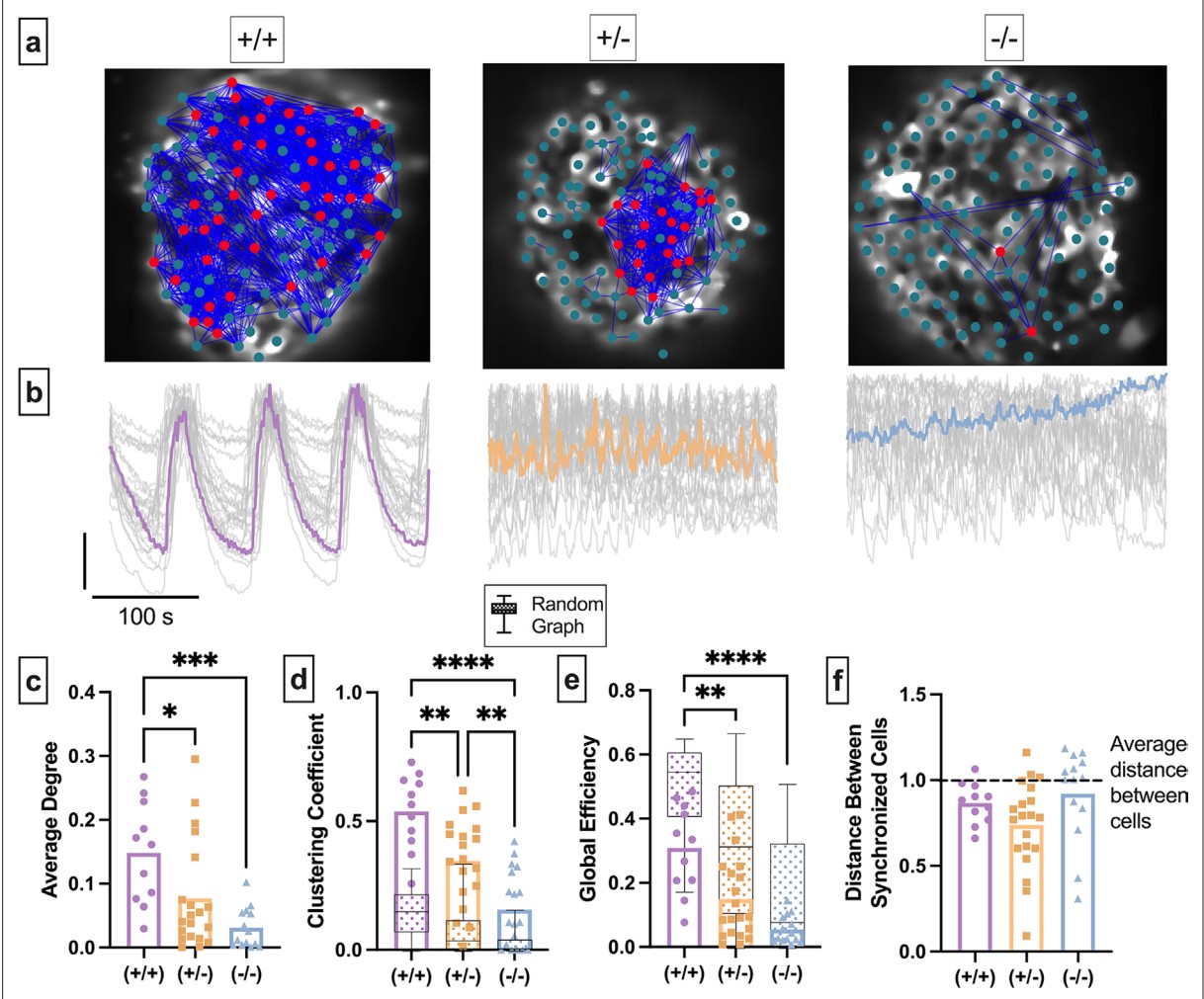

**Figure 6.** Experimental islet functional networks are influenced by changes in the structural network. (**a**) Representative images of Cx36$^{+/+}$ islet (left), heterozygous Cx36$^{+/-}$ islet (middle), and homozygous islet (right), overlaid with a synchronization network. Dot signifies a cell, blue lines represent functional network edges that connect synchronized cells. Red cells are β-cell hubs. (**b**) Oscillations in [Ca$^{2+}$] of corresponding islet in a. Gray lines represent time course for each cell, colored line represents mean islet time course. (**c**) Normalized degree of functional network between cell pairs for each islet in the Cx36$^{+/+}$, Cx36$^{+/-}$, Cx36$^{-/-}$ mice. (**d**) Clustering coefficient of the functional network for each islet in the Cx36$^{+/+}$, Cx36$^{+/-}$, Cx36$^{-/-}$ mice. Overlaid is the box and whisker plot of 1000 random networks per islet. (**e**) As in d for global efficiency. (**f**) Average distance between synchronized cells normalized to average distance between all cells in islet. Dashed line shows average distance between cells. Adjusted p-values: *p≤0.05, **p≤0.01, ***p≤0.001 For c–d, each dot represents an islet.

The online version of this article includes the following figure supplement(s) for figure 6:

**Figure supplement 1.** Alternative network metrics for calcium islets from Cx36KO (Cx36 knockout) mice.

A high clustering coefficient indicates that cells tend to synchronize with other cells that share some property, such as proximity or intrinsic dynamics. The clustering coefficient significantly decreased as gap junction coupling decreased (*Figure 6d*). To quantify whether the functional network topology was a result of random edge generation, we compared each metric with that determined from 1000 Erdos-Renyi random networks (*Watts and Strogatz, 1998*; *Gansterer et al., 2019*; *Stožer et al., 2013*; *Table 1*). If gap junctions are the property that gives the functional network a high clustering coefficient, then reducing gap junction coupling should result in a clustering coefficient that shows greater overlap with the random graph. For all levels of Cx36, the clustering coefficient of the islet functional network was greater than that of a random network (*Figure 6d*, box and whisker plot). Similar to our previous findings, this further suggests that something other than the structural network contributes to the functional network topology. *Global efficiency* is defined as the inverse of the average smallest number of nodes required to traverse between any cell pair (*Table 1*). High global

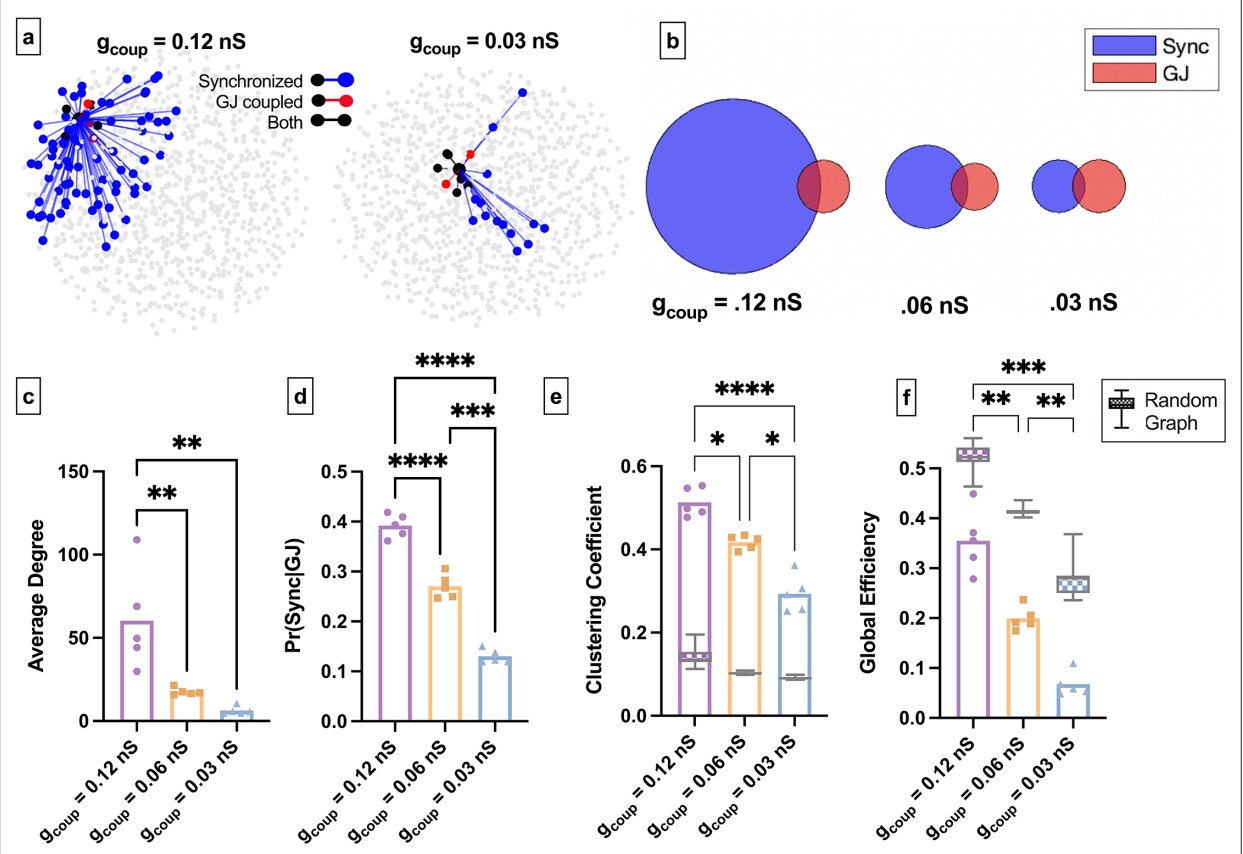

**Figure 7.** Simulated islet functional networks are influenced by changes in the structural network. (**a**) Functional connections and structural connections in simulated islets of hub cells for high and low gap junction conductance. Blue cells and edges indicate cells determined as 'synchronized' via functional network analysis. Red cells are GJ coupled. Black cells are both GJ coupled and synchronized. (**b**) Venn diagrams of the probability of synchronization and probability of gap junction with the probability of both synchronization and gap junction as the overlapped area. (**c**) Average degree in the functional network. (**d**) Probability that two cells are synchronized given they share a gap junction connection. (**e**) Clustering coefficient for 0.12nS, 0.06nS, 0.03nS , for each simulated islet. Overlaid is the box and whisker plot of 1000 random networks per simulated islet. (**f**) As in e for global efficiency for 0.12nS, 0.06nS, 0.03nS, for each simulated islet. Adjusted p-values: *p≤0.05 , **p≤0.01, ***p≤0.001. For c–f, each dot represents a simulated islet.

The online version of this article includes the following figure supplement(s) for figure 7:

**Figure supplement 1.** Alternative network metrics for simulated islet with gap junction coupling change.

efficiency can occur in a random network, or in a regular network (**Table 1**) with a few edges randomly moved (**Watts and Strogatz, 1998**). The global efficiency significantly decreased as gap junction coupling decreased (**Figure 6e**). For Cx36[+/+] islets, the global efficiency was less than that of a random network. As gap junction coupling decreased, the global efficiency showed a larger intersection with that of a random network. For Cx36[-/-] islets all measurements intersected with and thus could be explained by a random network. For each genotype, the distance between synchronized cells was less than the average distance between all cells (**Figure 6f**). These results suggest that the gap junction structural network contributes to whole islet global properties (global efficiency) of the functional network topology, but not immediate local properties (clustering), similar to our previous findings (**Figure 5**).

We next asked how changes to the structural network influence the islet functional network in the coupled Cha-Noma model. We decreased coupling conductance ($g_{coup}$) (**Figure 7a**). Similar to experimental findings, as average gap junction conductance decreased, the average degree and average correlation also decreased (**Figure 7c**, **Figure 7—figure supplement 1b**). Pr(Sync│GJ) also decreased significantly as gap junction conductance was decreased (**Figure 7b and d**). This indicates that as the gap junction conductance decreases, the probability that two structurally connected cells are also functionally connected decreases. We then quantified the functional network topology in the

coupled Cha-Noma model upon reduced gap junction conductance. The clustering coefficient ($C_{avg}$) progressively decreased with decreasing gap junction conductance and was always greater than that determined from a random network (*Figure 7e*). Global efficiency also progressively decreased with decreased gap junction conductance (*Figure 7f*) and was always less than that of a random network. These trends are similar to our experimental observations.

Overall, our findings from experimental and simulated islets both indicate that the gap junction structural network influences overall islet synchronization, as expected. However, while it influences the global functional network topology, it has less influence on the local functional network topology.

## Discussion

The islet of Langerhans is central to glucose homeostasis and dysfunction of the islet underlies diabetes. Heterogeneity among β-cells in the islet plays a key role in islet function (*Benninger and Kravets, 2022*; *Johnston et al., 2016*; *Salem et al., 2019*; *Kravets et al., 2022*). Functional network analysis enables the investigation of β-cell heterogeneity and how subpopulations of β-cells influence islet dynamics, over which there is currently a debate (*Dwulet et al., 2021*; *Satin et al., 2020*; *Rutter et al., 2020*; *Peercy and Sherman, 2022*). To correctly interpret the β-cell functional network and its implications in diabetes, we must understand what drives edges and subpopulations such as β-cell

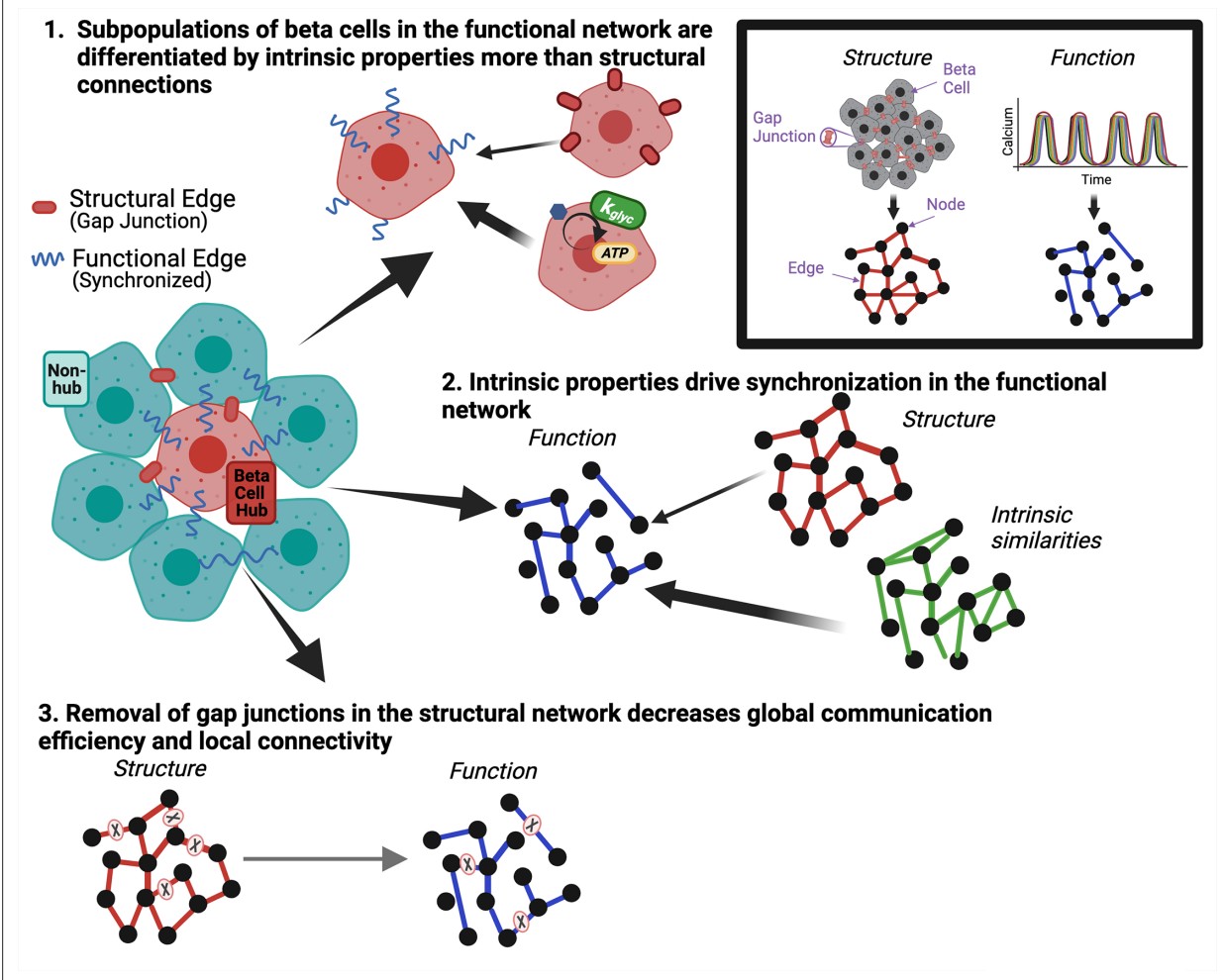

**Figure 8.** Graphical summary of results. Graphical summary of results from data. Inset shows how structural gap junctions are translated to structural network and synchronized calcium oscillations are translated to functional network. To answer question **1** we show that subpopulations of β-cells in the functional network are driven by intrinsic properties more than gap junction coupling (top). To answer question **2**, we show that intrinsic properties of the β-cells drive synchronization in the functional network (right). To answer question **3**, we show that removal of gap junctions in the structural network decreases global communication efficiency and local connectivity (bottom).

hubs. While it is known that gap junctions are essential for whole islet synchronization, it is unclear how the functional network relates to the gap junction structural network. Our aim was to investigate the relationship between functional networks, structural networks, and underlying β-cell dynamics in the pancreatic islets. We framed our study around three questions. (1) Are highly correlated subpopulations (β-cell hubs) that emerge from the β-cell functional network differentiated by their unique qualities with respect to the structural network or by intrinsic properties of the cells? (2) What does the islet functional network indicate about its underlying structure or intrinsic cell properties on an individual cell basis? (3) How do changes in the structural network affect the islet's functional network? *Figure 8* shows a graphical representation of the answers to these questions. By understanding the formation of islet functional networks, we can understand how β-cell subpopulations drive islet function. Knowledge about the role of β-cell subpopulations will also be useful when developing new treatments, such as identifying and preserving functional subpopulations in diabetes and generating stem cell-derived insulin-secreting cells for restoring insulin secretion.

## Highly connected β-cells are driven by intrinsic cell dynamics

Our computational and experimental results show a subpopulation of highly synchronized cells, known as β-cell hubs (*Johnston et al., 2016*; *Benninger and Hodson, 2018*). In the coupled Cha-Noma model of fast β-cell oscillations, the elevated rate of glucokinase $k_{glyc}$ (which indicates metabolic activity), was most strongly associated with β-cell hubs and duty cycle of the $[Ca^{2+}]$ oscillations (*Figure 1*). $g_{KATP}$ showed little association with duty cycle, which is consistent with the greater influence of ATP production (primarily controlled by $k_{glyc}$) and ATP-driven $K_{ATP}$ closure on membrane potential, compared to $K_{ATP}$ open-channel conductance (*Notary et al., 2016*). In the coupled IOM of slow β-cell oscillations, β-cell hubs were associated with both $g_{KATP}$ and gap junction conductance (*Figure 2*). In this case, the $g_{KATP}$ parameter showed a high association with oscillation duty cycle, as shown previously in uncoupled β-cell simulations (*Marinelli et al., 2022*). Thus, results from two independent models show that the cell intrinsic properties that influence oscillation duty cycle are associated with driving β-cell hubs, regardless of β-cell oscillation type. Notably an association between network degree and oscillation duty cycle has recently been demonstrated (*Šterk et al., 2023*; *Stožer et al., 2021*; *Gosak et al., 2022*).

A key difference between the two coupled models is the strong association between β-cell hubs and gap junction coupling in the coupled IOM and the weak association in the coupled Cha-Noma model. While there was an ~15% difference in $g_{coup}$ between hubs and non-hubs in the Cha-Noma model (compared to ~10% difference in $k_{glyc}$), this difference was subject to high variability and not observed for all thresholds used. This study represents one of the first uses of a coupled IOM to study β-cell heterogeneity. The Cha-Noma model, which has been used extensively to study β-cell heterogeneity (*Dwulet et al., 2021*; *Dwulet et al., 2019*; *Notary et al., 2016*; *Westacott et al., 2017*; *Silva et al., 2014*), is simulated with 1000 β-cells to roughly match experimental observations (*Pisania et al., 2010*; *Steiner et al., 2010*). However, we simulated the coupled IOM as a smaller islet model due to model stiffness and computational limits. When the coupled Cha-Noma model was simulated with a similarly small number of cells, gap junction conductance had a higher association with β-cell hubs (*Figure 2*). Therefore, we attribute the increased contribution of gap junction coupling to β-cell hubs in the coupled IOM to be partially influenced by islet size. We predict if the coupled IOM could be simulated with a larger number of cells, β-cell hubs would become more strongly associated with $g_{KATP}$ than gap junction coupling.

Our experimental measurements showed that β-cell hubs were associated with elevated metabolic activity, quantified by NAD(P)H fluorescence. In contrast, there was no association between β-cell hubs and gap junction permeability (*Figure 3*). Previous studies have shown that the functional network topology of β-cells is highly dependent on whether the β-cell oscillation is fast, slow, or plateau (*Zmazek et al., 2021*; *Stožer et al., 2022*). However, the relationship between β-cell hubs and gap junction permeability was not influenced by $[Ca^{2+}]$ oscillations type: slow, fast, and mixed. These results further illustrate how subpopulations of β-cell hubs in the functional network are likely emerge due to their intrinsic properties such as metabolic activity, rather than gap junction conductance. Importantly, we are unable to spatially resolve $K_{ATP}$ channel conductance or channel number, thus we cannot also exclude that β-cell hubs may show decreased $K_{ATP}$ conductance. Further, the variability that we observe comparing metabolic activity between hubs and non-hubs in both experiments and

simulated islets suggests other factors (including $K_{ATP}$ conductance) may potentially play some additional role in determining β-cell hubs.

Our findings also agree with previous evidence that β-cell hubs have elevated metabolic activity. Semi-quantitative immunofluorescence demonstrated elevated glucokinase protein in β-cell hubs, suggesting elevated glycolytic activity and increased accumulation of tetramethylrhodamine ethyl ester in β-cell hubs was indicative of increased oxidative phosphorylation (*Johnston et al., 2016*). However, evidence for elevated gap junction coupling in β-cell hubs was argued either on the basis of [$Ca^{2+}$] synchronization analysis which we show is not necessarily indicative of gap junction coupling (*Figures 1–5*) or a global knockdown of *GJD2* (which codes for Cx36) (*Johnston et al., 2016*) which will influence both β-cell hubs and non-hubs, as we also observe (*Figures 6–7*). Further supporting our findings here, prior experiments have shown that cells that are preferentially activated by optogenetic ChR2 stimulation, which indicates function, were more metabolically active, but not preferentially gap junction coupled (*Westacott et al., 2017*).

In answering our first question, we conclude that gap junctions are not the primary distinguishing factor in determining variations in oscillation synchronization between cells. Rather cell intrinsic properties including metabolic activity distinguish variations in synchronization between cells because of their influence on [$Ca^{2+}$] oscillation duty cycle.

## Functional network edges primarily reflect variations in intrinsic cellular properties

To compare the entire islet functional network with its structural network (instead of hubs and non-hubs in question 1), we analyzed the coupled Cha-Noma model, and generated conditional probabilities that quantify the relationship between structural and functional networks at an individual cell basis (*Figure 4*). We found the probability of a gap junction connection given a synchronized cell pair (Pr(*GJ|Sync*)) was <5%, indicating that high [$Ca^{2+}$] synchronization rarely implies a gap junction connection. However, the probability that two synchronized cells had similar metabolic activity was significantly larger (~20%). *Therefore, the functional network primarily reflects variations in cellular metabolic activity rather than structural gap junction connections.*

Importantly the overlap in cellular metabolic activity with the functional network refers to cells sharing some intrinsic metabolic properties ($k_{glyc}$) that drives duty cycle and functional network edges. It has been suggested that β-cells may be metabolically coupled, where metabolites diffuse between cells via gap junctions (*Rao and Rizzo, 2020*; *Tsaneva-Atanasova et al., 2006*). However, our findings regarding metabolic activity or intrinsic dynamics driving the functional network are not referring to metabolic coupling, but rather the fact that intrinsically similar cells will appear more synchronized, and therefore will be more likely to be connected in the functional network.

Surprisingly, we found that gap junctions do not guarantee that [$Ca^{2+}$] synchronization will be higher than the functional network threshold ($R_{th}$), as previously assumed (*Markovič et al., 2015*). The probability of high synchronization given gap junction connections (Pr(*Sync|GJ*)) was only ~40% (*Figure 4*). When threshold is reduced, Pr(*Sync|GJ*) increases to almost 100% (*Figure 1—figure supplement 1e*) but the functional degree distribution no longer appears scale-free-like, as indicated by previous studies (*Stožer et al., 2013*; *Johnston et al., 2016*). Gap junctions facilitate ion passage between β-cells and therefore play a primary role in synchronizing electrical oscillations across the islet (*Figure 5*). However, they are not the only factor contributing to the high synchronization seen between some β-cell pairs. The cell membrane potential is determined by cell metabolic activity, $K_{ATP}$ conductance, and other ion channels. The membrane potential of 2 cells determines the electrochemical gradient along the gap junction, and thus the coupling current which drives synchronization (*Dwulet et al., 2021*; *Plonsey et al., 2007*). As such, high metabolic activity can drive a high coupling current. Furthermore, cells with similar intrinsic characteristics, such as metabolic activity, require less electrical coupling to be highly synchronized. *Thus, while gap junctions are important for the overall synchronization of islet [$Ca^{2+}$] dynamics, additional intrinsic β-cell factors are necessary for very strong synchronization between proximal β-cells.*

## Perturbing the gap junction structural network altered the islet functional network topology

Changes in gap junction coupling and [Ca$^{2+}$] oscillation synchronization are observed in conditions associated with diabetes. Impaired [Ca$^{2+}$] oscillation synchronization (representing a disrupted functional network) and disrupted pulsatile insulin secretion have been observed in islets from rodent models of type 2 diabetes (*Pørksen et al., 2002*; *Satin et al., 2015*; *Corezola do Amaral et al., 2020*), type 1 diabetes (*Pørksen et al., 2002*; *O'Meara et al., 1995*), human subjects with type 2 diabetes (*St. Clair et al., 2020*) or obesity (*Hodson et al., 2013*), and when exposed to glucolipotoxicity, or pro-inflammatory cytokines (*Hodson et al., 2013*; *Farnsworth et al., 2016*). Similarly, altered gap junction coupling (representing a disrupted structural network) has been observed in islets from mouse models of type 2 diabetes (*St. Clair et al., 2020*; *Corezola do Amaral et al., 2020*), type 1 diabetes (*Farnsworth et al., 2022*), prediabetes (*Carvalho et al., 2012*), and exposure to hyperglycemia or pro-inflammatory cytokines (*Farnsworth et al., 2016*; *Haefliger et al., 2013*). When gap junction coupling and [Ca$^{2+}$] synchronization are perturbed upon a genetic deletion of Cx36, first-phase insulin secretion and pulsatile second-phase secretion are diminished, causing glucose intolerance. Therefore, it is important to understand the relationship between altered [Ca$^{2+}$] oscillation synchronization and gap junctions coupling.

Our computational and experimental results show that as structural edges were removed by reducing gap junction conductance, the functional network degree, efficiency, and clustering decreased (*Figures 6 and 7*). This implies that the islet loses efficient signal propagation and the ability to synchronize. This in part explains decreased islet function in diabetes. However, clustering could not be explained by randomness (*Figures 6 and 7*). If gap junctions were solely responsible for variations in islet synchronization, a decrease in gap junction coupling would cause the functional network to appear more random. Instead, our findings indicate that β-cells are preferentially, not randomly, synchronized even in the presence of low gap junction coupling. This is consistent with synchronization between cell pairs being driven by intrinsic cellular dynamics, such as metabolic activity. Thus, while removal of gap junctions is detrimental to overall islet function, changes in synchronization are not directly indicative of only changes in structural connections. Perturbations in cellular metabolic activity, which occurs in diabetes (*Ostenson et al., 1993*), will also significantly influence islet synchronization. Additionally, our results indicate that the relationship between the structural and functional network is not strongly influenced by β-cell oscillation type (*Figure 3—figure supplement 1*), therefore, fast, slow, and mixed oscillations should show similar changes with decreased gap junction coupling.

## Potential limitations

Our computational models are based on careful physiological recordings and have been previously validated (*Dwulet et al., 2019*; *Hraha et al., 2014*; *Westacott et al., 2017*; *Bertram et al., 2018*). Nevertheless, they will still be limited in describing β-cell dynamics. Because our results are internally derived, where the functional and structural networks were obtained from the same model, they are not dependent on the models' ability to reflect islet dynamics. This fact also allows our methods to translate to any oscillatory network with similar characteristics – this is reflected by our use of two independent coupled models. The choice of threshold ($R_{th}$) is influential in creating the functional network and should be carefully considered. We used thresholds to match that of the prior experimental results (*Stožer et al., 2013*; *Johnston et al., 2016*; *Korošak et al., 2021*). The computational thresholds were higher in these prior experiments because the computational model did not include noise, so that synchronization was more acute. However, the trends in our results for the Cha-Noma model and experiment were consistent across thresholds and hub definition (*Figure 1—figure supplements 1 and 2*, *Figure 3—figure supplements 1 and 2*). Within the coupled IOM, a very high threshold was required to generate a degree distribution that matched the coupled Cha-Noma model results. This was likely due to the small islet size, which was limited by model instability. Nevertheless, consistent results were found in terms of parameters that influence duty cycle associating with β-cell hubs.

FRAP measurements of gap junction conductance are likely less sensitive than gold-standard patch clamping, but it allows measurements in multiple cells in situ and subsequent calcium imaging, which was central to our study. Additionally, we cannot exclude that the Cx36 global deletion may influence β-cell specification. Thus, generating effective inducible and conditional Cx36 knockout models are

needed. However, studies have shown no significant change in NAD(P)H response and islet architecture in Cx36 knockout islets (*Benninger et al., 2011*). Dissociated β-cells of both Cx36 knockout and WT islets also show similar insulin secretion responses, indicating a Cx36 knockout islets has no significant defect in the intrinsic ability of a β-cell to release insulin.

Dynamic changes in the distribution of gap junction coupling can occur over time (*Miranda et al., 2022a*), which is not accounted for in our computational models. However, since our experimental results indicate that β-cell hubs are more influenced by intrinsic activity, we do not predict that changes in the distribution of gap junction coupling over time would substantially impact β-cell hubs.

Due to propagating Ca²⁺ waves across the islet, there is a phase lag between [Ca²⁺] oscillation that is small for β-cells in close proximity and greater for β-cells far apart. A large phase lag will decrease the apparent [Ca²⁺] oscillation synchronization, thus influencing the functional network. Future work will be needed to examine 'wave initiator' like cells that lie at the origin of propagating Ca²⁺ waves, or cells located at the wave end, and determine their position and influence on the islet functional network. Indeed, prior work has shown that cells at the wave end likely influence the overall [Ca²⁺] oscillation dynamics (*Dwulet et al., 2021*).

The islet contains α-cells, δ-cells, and other cell types which can influence β-cell dynamics via paracrine mechanisms (*Henquin, 2021*; *Moede et al., 2020*; *van der Meulen et al., 2015*). Our study only analyzes β-cells, as performed in other network studies which we compare (*Stožer et al., 2013*; *Johnston et al., 2016*; *Lei et al., 2018*). However, we cannot exclude long-range structural projections from δ-cells (*Arrojo E Drigo et al., 2019*; *Briant et al., 2018*), and that δ-cells may be gap junction coupled (*Miranda et al., 2022b*). Additionally, β-cell oscillation types (slow, fast, or mixed) may be directly related to the number of α-cells present in the islet (*Ren et al., 2022*). Future analysis should include other cell types, which may provide valuable insight on how their interactions can shape the overall islet response. This is particularly important when translating our findings to human islets because α-cells, δ-cells, and β-cells are known to be more mixed in human islets (*Kim et al., 2009*), and there is experimental evidence that human islet functional networks may differ from mouse functional networks (*Gosak et al., 2022*).

## Overall summary

While it is well known that gap junctions allow for whole islet synchronization (*Benninger et al., 2008*), it is not clear whether the functional network directly reflects gap junction structural network or if the functional network is influenced by intrinsic β-cell characteristics. We investigated the relationship between the synchronization-based functional network, the gap junction structural network, and β-cell intrinsic properties in the islets of Langerhans. We concluded with three major points. *First*, highly connected subpopulations of β-cells, including β-cell hubs, are defined by intrinsic properties more than gap junctions. *Second*, the functional network primarily reflects variations in intrinsic cellular properties. *Third,* removal of edges in the structural network upon decreasing gap junction conductance decreases global communication efficiency and local connectivity and causes the functional network to become sparser. Our results also show that forming conclusions about structure using the functional network should be done with caution. Broadly, these results provide insight into the relationship between function and structure in biological networks and the understanding that synchronization can give insight into cellular properties that drive excitability of the cell can be useful in many systems.

## Methods
### Calcium imaging, FRAP, and NAD(P)H
#### Animal care
Male and female mice were used under protocols approved by the University of Colorado Institutional Animal Care and Use Committee. β-Cell-specific GCaMP6s expression (β-GCaMP6s) was achieved through crossing a MIP-CreER (The Jackson Laboratory) and a GCaMP6s line (The Jackson Laboratory) (*Kravets et al., 2022*). Genotype was verified through qPCR (Transetyx, Memphis, TN, USA). Mice were held in a temperature-controlled environment with a 12 hr light/dark cycle and given continuous access to food and water. CreER-mediated recombination was induced by 5 daily doses of tamoxifen (50 mg/kg bw in corn oil) delivered IP.

## Islet isolation and culture

Islets were isolated from mice under ketamine/xylazine anesthesia (80 and 16 mg/kg) by collagenase delivery into the pancreas via injection into the bile duct. The collagenase-inflated pancreas was surgically removed and digested. Islets were handpicked and planted into the glass-bottom dishes (MatTek) using CellTak cell tissue adhesive (Sigma-Aldrich). Islets were cultured in RPMI medium (Corning, Tewksbury, MA, USA) containing 10% fetal bovine serum, 100 U/mL penicillin, and 100 mg/mL streptomycin. Islets were incubated at 37°C, 5% $CO_2$ for 24–72 hr before imaging.

## Imaging

An hour prior to imaging nutrition media from the isolated islets was replaced by an imaging solution (125 mM NaCl, 5.7 mM KCl, 2.5 mM $CaCl_2$, 1.2 mM $MgCl_2$, 10 mM HEPES, and 0.1% BSA, pH 7.4) containing 2 mM glucose. During imaging the glucose level was raised to 11 mM. Islets were imaged using either an LSM780 system (Carl Zeiss, Oberkochen, Germany) with a 40×1.2 NA objective or with an LSM800 system (Carl Zeiss) with 20×0.8 NA PlanApochromat objective, or a 40×1.2 NA objective, with samples held at 37°C.

For [$Ca^{2+}$] measurements GCaMP6s fluorescence was excited using a 488 nm laser. Images were acquired at 1 frame/s at 10–20 μm depth from the surface of the islet. Glucose was elevated 3 min after the start of recording, unless stated otherwise.

NAD(P)H autofluorescence and [$Ca^{2+}$] dynamics were performed in the same z-position within the islet. NADH(P)H autofluorescence was imaged under two-photon excitation using a tunable mode-locked Tisapphire laser (Chameleon; Coherent, Santa Clara, CA, USA) set to 710 nm. Fluorescence emission was detected at 400–450 nm using the internal detector. Z-stacks of 6–7 images were acquired spanning a depth of 5 μm. First NAD(P)H autofluorescence (1 z-stack) was recorded at 2 mM glucose, then the [$Ca^{2+}$] dynamics was recorded at 2 mM and during transition to 11 mM glucose (~30 min). After measuring the oscillatory [$Ca^{2+}$] time course, NAD(P)H autofluorescence (1 z-stack) was recorded at 11 mM glucose.

Cx36 gap junction permeability and [$Ca^{2+}$] dynamics were performed in the same z-position within the islet, with gap junction permeability measured using FRAP, as previously described. After [$Ca^{2+}$] imaging, islets were loaded with 12 mM Rhodamine-123 for 30 min at 37°C in imaging solution. Islets were then washed and FRAP performed at 11 mM glucose at room temperature. Room temperature was used because at this temperature the Rhodamine-123 travels between the cells only through the Cx36 gap junctions, versus at 37°C it can permeate a cell membrane. Rhodamine-123 was excited using a 488 nm laser line, and fluorescence emission was detected at 500–580 nm. Three baseline intensity images were initially recorded. A region of interest was then photobleached achieving, on average, a 50% decrease in fluorescence, and images were then acquired every 5–15 s for 15–30 min.

## Analysis of [$Ca^{2+}$] dynamics

Pearson-product-based network analysis presented in *Figure 2* was performed as previously reported (*Stožer et al., 2013*). [$Ca^{2+}$] time courses were analyzed during the second-phase [$Ca^{2+}$] response when the slow calcium wave was established. For fast oscillations, 400–500 s of [$Ca^{2+}$] response was analyzed, matching the simulation time. For slow oscillations, 741–1000 s of [$Ca^{2+}$] response was analyzed. This time period was chosen so at least 3 oscillations were completed. In mixed oscillations, 858–1000 s of [$Ca^{2+}$] response was analyzed.

The Pearson product for each cell pair islet was calculated over each time point, and the time-average values were computed to construct a correlation matrix. An adjacency matrix was calculated by applying a threshold to the correlation matrix. The same threshold of 0.9 was applied to all islets. All cell pairs with a non-zero values in the adjacency matrix were considered to have a functional edge. The percent of edges was calculated with respect to the maximum number of edges per cell in each individual islet. For example, if a most connected cell possessed max = 10 edges, and other cells had 1, 3, …, 7 edges – then the % were: 10%, 30%, …, 70%.

## Prior calcium imaging first presented in (*Benninger et al., 2008*)

### Islet isolation

Islets were isolated as described in *Scharp et al., 1973*, and *Stefan et al., 1987*, and maintained in Roswell Park Memorial. Institute medium containing 10% fetal bovine serum, 11 mM glucose at 37°C under humidified 5% $CO_2$ for 24–48 hr before imaging.

### Imaging islets

Isolated islets were stained with 4 mM Fluo-4 AM (Invitrogen, Carlsbad, CA, USA) in imaging medium (125 mM NaCl, 5.7 mM KCl, 2.5 $CaCl_2$, 1.2 mM $MgCl_2$, 10 mM HEPES, 2 mM glucose, 0.1% bovine serum albumin, pH 7.4) at room temperature for 1–3 hr before imaging. Islets were imaged in a polydimethylsiloxane microfluidic device, the fabrication of which has been previously described in Rocheleau et al. which holds the islet stable for imaging and allows rapid reagent change, such as varying glucose stimulation or adding gap junction inhibitors. Fluo-4 fluorescence is imaged 15 min after a step increase in glucose from low (2 mM) to high (11 mM). High-speed imaging is performed on an LSM5Live with a 203 0.8 NA Fluar Objective (Zeiss, Jena, Germany) using a 488 nm diode laser for excitation and a 495 nm long-pass filter to detect fluorescence emission. The microfluidic device is held on the microscope stage in a humidified temperature-controlled chamber, maintained at 37°C. Images were acquired at a rate of 4–6 frames/s, with average powers at the sample being minimized to 200 mW/cm².

### Analysis of $Ca^{2+}$ imaging data

We present data from 11 islets from 6 WT mice, 11 islets from 11 heterozygous $Cx36^{+/-}$ knockout mice, and 14 islets from 3 homozygous $Cx36^{-/-}$ knockout mice. We extracted cell calcium dynamics by visually identifying and circling all cells islet. We assumed that the pixels within a cell should be well coordinated, so we removed any pixels whose dynamics were not within 5–10 STD of the average. This usually resulted in the removal of 1–5 pixels on the edge of the cell boundary.

### Threshold

Previous studies have shown that WT islets should have a degree distribution that is roughly linear when plotted on a log-log plot and average degree of either 8 or between 5 and 15 (*Stožer et al., 2013*; *Johnston et al., 2016*; *Zmazek et al., 2021*; *Korošak et al., 2021*). To determine $R_{th}$ that best satisfied both of these findings, we utilized constrained optimization MATLAB (MathWorks Inc, Natick, MA, USA) algorithm fminsearchbnd (*D'Errico, 2023*) to find the optimal $R_{th}$ that maximized the goodness of fit to a power law distribution, while forcing $5 \leq k_{avg} \leq 15$ for each WT islet constrained optimization. The average optimal threshold ($R_{th} = 0.90$).

### Average distance between connected cells

The average distance between connected cells calculated the total number of pixels between the center of two connected cells. The average distance was expressed as the normalized distance between connected cells and the average distance between all cells in the islet to control for image and islet size.

## Computational model

### Coupled Cha-Noma model

This ordinary differential equation model has been described previously (*Dwulet et al., 2021*) and validated on experimental studies (*Dwulet et al., 2019*; *Notary et al., 2016*; *Hraha et al., 2014*; *Westacott et al., 2017*). This model has been shown to accurately describe experimental findings concerning spatial-temporal dynamics (*Westacott et al., 2017*), the relationship between electrical coupling and metabolic heterogeneity (*Dwulet et al., 2019*), the relationship between electrical heterogeneity and electrical activity (*Hraha et al., 2014*), and the influence of excitability parameters such as rate of glucose metabolism ($k_{glyc}$) and $K_{ATP}$ channel opening kinetics on diabetes mutations (*Dwulet et al., 2019*; *Notary et al., 2016*). It is based on a single β-cell electrophysiological model (*Cha et al., 2011*), in which the membrane potential ($V_i$) for β-cells$_i$ is solved for each time step using *Equation 1a*. We created a 1000 β-cell network and electrically coupled any cell pairs within a small

distance from each other. We chose 1000 β-cell because most species (including human and mouse) contain on average 1000 β-cells in an islet (*Pisania et al., 2010*) and it is the number which has been validated experimentally. The coupling current, $I_{coup}$, is determined by the difference in voltage between the cell pair and the average coupling conductance ($g_{coup}$) between them (*Equation 1b*). All code was written in C++ and run on the SUMMIT supercomputer (University of Colorado Boulder). All simulations are run at 11 mMol glucose. The membrane potential ($V_i$) for β-cell *i* is solved for using the ODE:

$$Cm\frac{dV_i}{dt} = I_{Cav} + I_{TRPM} + I_{SOC} + I_{bNSC} + I_{KDr} + I_{KCa(SK)} + I_{K_{ATP}} + I_{NaCa} + I_{PMCA} + I_{NaCa} + I_{coup} \quad (1a)$$

$$I_{coup} = \sum_i g_{coup}^{ij} \left(V_i - V_j\right) \quad (1b)$$

The number of gap junction connections was given by a normal distribution mean 5.25, standard deviation 1.6, with the maximum of 12 and a minimum of 1 gap junction connections per cell. Cellular heterogeneity was introduced by assigning randomized metabolic and electrical parameter values to each cell based on their distributions previously determined by experimental studies. For example, glucokinase ($k_{glyc}$), the rate-limiting step in glycolysis, was assigned using a normal distribution with mean $1.26*10^{-04}$ s$^{-1}$ and standard deviation $3.15*10^{-05}$ s$^{-1}$. Conductance of the KATP channel ($g_{KATP}$) was given by a normal distribution with mean 2.31 pA/mV and standard deviation 0.57 pA/mV. The coupling conductance parameter of a cell ($g_{coup}^j$) was assigned using a gamma distribution with k=4, θ=4 and then shifted so the islet average was $g_{coup}^j$ =0.12 nS. This resulted in average $\sigma_{coup}$ =0.12 nS and standard deviation of 0.06 nS. We ran the simulation for 500 s and only the second phase was analyzed, in accordance with *Johnston et al., 2016*. All the results presented are based on five different islets, including all 1000 cells, with randomized parameter values. To explore the effects of coupling conductance on the functional network, we altered islet average coupling conductance of a single gap junction pair to $g_{coup}^j$ =0.06 nS and $g_{coup}^{ij}$ =0.03 nS, and then assigned each cell a randomized $g_{coup}^j$ based on the new target average. The total gap junction conductance ($g_{coup}$) for any cell was calculated as the sum of the average conductance between the cell of interest and any cells it shares a gap junction connection with:

$$g_{coup} = \sum_{i=1}^{m} \frac{1}{2} \left(g_{coup}^i + g_{coup}^j\right) \quad (2)$$

## The coupled IOM

The ordinary differential equations for the IOM have been extensively described in previous works (*Marinelli et al., 2021*; *Marinelli et al., 2022*). The model simulates the dynamics within a single β-cell and it has been shown to be able to account for key experimental findings (*Bertram et al., 2018*; *Marinelli et al., 2018*), including both slow and fast bursting regimes. Allowing this achievement is the interaction of the three modules that make up the model: one module to describe the cellular electrical activity, a second to describe the glycolysis, and a third to describe the mitochondrial metabolism. The change in the membrane potential ($V_i$) for β-cell$_i$ is described by *Equation 3*.

$$Cm\frac{dV_i}{dt} = I_{Ca} + I_{K(Ca)} + I_{K(ATP)} + I_K \quad (3)$$

Due to limitations of computational power and model stability, we created a 260 β-cell network (instead of the 1000 cells simulated with the coupled Cha-Noma model) where cells we coupled based on their proximity and through the coupling conductance, $I_{coup}$, defined in *Equation 1b*. The differential equations were integrated numerically using MATLAB and all simulations are run at 11 mMol glucose.

In this simulation, the average number of gap junction connections per cell was 6.01 with a standard deviation of 2.66, and with a maximum of 14 and a minimum of 1 gap junction connection per cell. Similar to the computation with the Cha-Noma model, cellular heterogeneity was achieved by allowing key metabolic and electrical parameters to range within a distribution. More precisely, the coupling conductance ($g_{coup}$) is assumed to be normally distributed with mean 1 pS and a standard deviation 0.5. We then assign the maximum rate through the glucokinase reaction ($k_{glyc}$, referred to

previously as $v_{GK}$; *Marinelli et al., 2021*; *Marinelli et al., 2022*) using a normal distribution with a mean 0.0037 µM/ms and standard deviation 0.0015 µM/ms, and extract the values for the maximum conductance of the KATP channel ($g_{KATP}$) from a normal distribution with mean 19,700 pS and standard deviation 3940 pS. Once again, the total gap junction conductance ($g_{coup}$) for any cell was calculated as described in *Equation 2*.

## Network analysis

### Creating functional network from Cha-Noma simulated islet

The methodology was based on that previously defined in *Stožer et al., 2013*. First, the correlation coefficient between each cell was calculated using corr() function in MATLAB, which follows the equation:

$$R_{ij} = \frac{\sum \left[ x_i\left(t\right) - \bar{x}_i \right] \left[ x_j\left(t\right) - \bar{x}_j \right]}{\sqrt{\sum [x_i\left(t\right) - \bar{x}_i]^2 \sum [x_i\left(t\right) - \bar{x}_i]^2}}$$

Next, a threshold ($R_{th}$) was defined to determine whether each cell pair is 'synchronized' or 'not synchronized'. For computational experiments, the threshold was chosen such that the network roughly followed a power law distribution, as predicted in *Stožer et al., 2013*; *Johnston et al., 2016*. Unless otherwise noted, $R_{th} = 0.9995$ for computational analysis.

### Creating functional network from IOM simulated islet

To create the functional network from the IOM simulated islet, we repeated steps for the Cha-Noma model. However, due to small islet size and model stiffness, we were unable to achieve strong heterogeneity in the calcium oscillations (*Figure 2a*), requiring a threshold of $R_{th} = 0.999999999$ to obtain a scale-free-like distribution (*Figure 2b and c*).

### Creating the metabolic network

Because gap junctions enforce localization onto the analysis, we looked at cell pairs whose Pythagorean distance was ≤15% within the islet. Within this sample space, we looked at cell pairs whose average $k_{glyc}$ was more similar than the islet average (*Figure 4b*).

We chose these parameters such that the Pr(*Sync*|*GJ*)=Pr(*Sync*|*Met*), which acts as a control allowing us to directly compare Pr(*GJ*|*Sync*) to Pr(*Met*|*Sync*).

### Average degree

The degree $k_i$ was calculated by counting the number of connections for cell $i$ and averaging $k$ over all cells in the islet, then normalized to the islet size to remove any size dependence ($k_{avg}$ / n, where n = islet size).

### Degree distribution

The degree distribution was calculated by first calculating the degree of each cell by taking the column sum of the adjacency matrix. Then each cell degree was normalized by dividing by the maximum degree of the islet. The histogram was then calculated using GraphPad Prism.

### Hub identification

In accordance with *Johnston et al., 2016*, any cell with more than 60% islet, calculated by the degree distribution, was considered a hub cell.

### Probabilities

We first created the functional network and structural (GJ-related probabilities) or metabolic network (metabolism). We then calculated the probability that two cells were synchronized by:

$$\Pr\left(sync\right) = \frac{\frac{m_{sync}}{(n-1)*n}}{2}$$

where $m_{sync}$ = number of edges in the synchronized network, and $n$ = number of nodes. Similarly, we calculated the probability that two cells were either gap junction coupled or metabolically related using

$$\Pr\left(GJ \; or \; k_{glyc}\right) = \frac{\frac{m_{GJ \; or \; k_{glyc}}}{(n-1)*n}}{2}$$

where $m_{GJ \; or \; k_{glyc}}$ is the number of edges in the gap junction or metabolic network.

To find the probability that a cell pair was both synchronized and GJ or metabolically connected, we calculated using equation:

$$\Pr\left(both\right) = \frac{\frac{m_{both}}{(n-1)*n}}{2}$$

where $m_{both}$ is any edge that exists in both matrices.

Finally, we calculated conditional probabilities by:

$$\Pr\left(sync|GJ\right) = \frac{\Pr\left(both\right)}{\Pr\left(GJ\right)}$$

$$\Pr\left(GJ|sync\right) = \frac{\Pr\left(both\right)}{\Pr\left(sync\right)}$$

These quantities were calculated separately for each simulated islet and then averaged.

## Duty cycle

We defined duty cycle as the proportion of time [Ca$^{2+}$] was elevated above 50% of its maximum value.

## Network topology analysis

### Shortest weighted path length

The gap junction network was weighted using the inverse of $g_{coup}$ or $k_{glyc}$ between cell $i$ and cell $j$ : $W_{ij} = \frac{1}{\frac{1}{2}\left(g_{coupi}+g_{coupj}\right)}$ or $W_{ij} = \frac{1}{\frac{1}{2}\left(k_{glyci}+k_{glycj}\right)}$ . This is done because the shortest path length algorithm views weights as 'resistance' and finds the path of least resistance. The shortest weighted path between every cell was calculated using Johnson's algorithm (*Johnson, 1977*). Cell pairs were then categorized as synchronized or not synchronized if their correlation coefficient was $<R_{th}$ . The average for synchronized and non-synchronized cells was calculated over each islet for each distance. To normalize over distance, each data point was divided by the average of the non-synchronized and synchronized islets for the given distance.

### Clustering coefficient

The clustering coefficient represents the 'cliquishness' of the network. This is defined by *Stožer et al., 2013*, as the 'number of existing connections between all neighbors of a node, divided by the number of all possible connections between them'. This was calculated by making a subgraph of each cell's connections and counting the number of connections between those cells. For example, if A is connected to B, C, and D, and B and C are connected but D is not connected to any other cell (see matrix). Then the clustering for cell A is $\frac{2}{3*2} = \frac{1}{3}$ . Each node is assigned a clustering coefficient $C$ such that:

| Subgraph – A | B | C | D |
|---|---|---|---|
| B |  | 1 | 0 |
| C | 1 |  | 0 |
| D | 0 | 0 |  |

The average clustering coefficient is $C_{avg} = \frac{1}{n}\sum_{i}^{n} C_i$ .

## Average shortest path length

Shortest path length was calculated with MATLAB function graphallshortestpaths(). This function uses the Johnson's algorithm (*Johnson, 1977*) to find the shortest path between every pair of cell islet. For example, the path length between cell $i$ and cell $j$ is 1 if they are directly synchronized, or 2 if cell $i$ is not synchronized with cell $j$ but each is synchronized with cell $k$ . To compensate for highly sparse network, any non-connected node was given a characteristic path length of $n+1$. Finally, the characteristic path length ($L$), or average path length, was expressed as the sum of all path lengths normalized to total possible connections (size*(size-1)).

The normalized average shortest path length (Supplemental 2d) is therefore calculated as $L_{avg} = \frac{1}{n} \frac{1}{n} \sum_{i=1}^{n} \sum_{j=1}^{n} L_{ij}$. To compensate for any non-connected cell, whose path length $L_{ix} = \infty$, where $x$ is any cell islet, we set the path length for non-connected cells to $L_{ix} \to n+1$, where $n$ is the number of cells islet.

## Global efficiency

The global efficiency is related to the inverse of global path length (*Latora and Marchiori, 2001*).

$$E_{global} = \frac{1}{Islet_{size} * (Islet_{size} - 1)} \sum_{jk} \frac{1}{L_{jk}}$$

length using $E_{global} = \frac{1}{n(n-1)} \sum_{i=1}^{n} \sum_{j=1}^{n} \frac{1}{L_{ij}}$ . Because $E_{global} = 0$ for a non-connected cell, the disconnected network is naturally compensated for.

## Random networks

Random networks were created using an Erodos Renyi approach (*Lakens, 2013*). For each islet, we created 1000 random networks with equal number of nodes, edges, and average degree as the islet of interest. The probability that two cells were connected was given by p = $\frac{k_{avg}}{n}$. For each cell pair within the islet, a random number generator created a number from 0 to 1. If that number was below p, whose cells were connected by an edge.

## Significance testing

Effect sizes were calculated using Cohen's $d_z = t/\sqrt{n}$ which is appropriate for paired samples (*Lakens, 2013*), where n=5.

Significance tests are done in Prism (GraphPad). $\alpha$ values are set to 0.5 unless otherwise mentioned.

*Figure 1e–g*, *Figure 2d–f and h–j*, *Figure 4e, f* and *Figure 5e, h* are paired two-tailed t-tests.

*Figure 1h*, *Figure 2g*, *Figure 6c–f*, and *Figure 7c–f* are repeated measures paired one-way ANOVA with multiple comparison using Tukey's multiple comparison.

*Figure 3f, i* are unpaired two-tailed t-tests.

*Figure 3g and j* are linear regressions.

*Figure 1—figure supplement 1b* is repeated measures paired one-way ANOVA with multiple comparison using Tukey's multiple comparison.

*Figure 1—figure supplement 2a-c*, *Figure 2—figure supplement 1a-f*, *Figure 2—figure supplement 2a-c*, *Figure 3—figure supplement 1d and g*, *Figure 3—figure supplement 2a-b*, *Figure 6—figure supplement 1b*, *Figure 7—figure supplement 1b* are two-tailed t-tests.

*Figure 5—figure supplement 1b*, *Figure 5—figure supplement 2b* are multiple paired t-tests with Bonferroni-Dunn adjustment. Since there were 15 cell distances, we set the significance threshold $\alpha$=0.003. For convenience, we present asterisks next to significant p-values.

## Acknowledgements

Richard KP Benninger (University of Colorado) is the guarantor of this work and, as such, had full access to all the data in the study and takes responsibility for the integrity of the data and the accuracy of the data analysis. We thank David W Piston (Washington University St Louis) with whom data presented in *Figure 6* was previously acquired. We thank Aaron Clauset and Mark Husing for helpful conversations and advice. All authors acknowledge that no conflict of interest exists. National Institutes of

Health (NIH) grant R01 DK102950 (RKPB) National Institutes of Health (NIH) grant R01 DK106412 (RKPB) National Science Foundation (NSF) Graduate Research Fellowship DGE-1938058_Briggs (JKB) Juvenile Diabetes Research Foundation (JDRF) grant 3-PDF-2019-741-A-N (VK) Beckman Research Institute-City of Hope (HIRN) grant UC24 DK104162 (VK) Burroughs Wellcome Fund – Careers at Scientific Interfaces Project Number 25B1756 (VK) National Institutes of Health (NIH) grant F31 DK126360 (JMD) National Institutes of Health (NIH) grant LM012734 (DJA) University of Birmingham Dynamic Investment Fund (IM). The authors are grateful for the utilization of the SUMMIT supercomputer from the University of Colorado Boulder Research Computing Group, which is supported by the National Science Foundation (awards ACI-1532235 and ACI-1532236), the University of Colorado Boulder, and Colorado State University. Microscopy use was supported in part by NIH grant P30 DK116073 and the University of Colorado Neurotechnology center.

## Additional information

### Funding

| Funder | Grant reference number | Author |
|---|---|---|
| National Institutes of Health | R01 DK102950 | Richard KP Benninger |
| National Institutes of Health | R01 DK106412 | Richard KP Benninger |
| National Science Foundation | Graduate Research Fellowship DGE-1938058_Briggs | Jennifer K Briggs |
| Juvenile Diabetes Research Foundation United States of America | 3-PDF-2019-741-A-N | Vira Kravets |
| Beckman Research Institute, City of Hope | UC24 DK104162 | Vira Kravets |
| Burroughs Wellcome Fund | 25B1756 | Vira Kravets |
| National Institutes of Health | DK126360 | JaeAnn M Dwulet |
| National Institutes of Health | LM012734 | David J Albers |
| University of Birmingham | Dynamic Investment Fund | Isabella Marinelli |

The funders had no role in study design, data collection and interpretation, or the decision to submit the work for publication.

### Author contributions

Jennifer K Briggs, Conceptualization, Data curation, Software, Formal analysis, Funding acquisition, Validation, Investigation, Visualization, Methodology, Writing – original draft, Writing – review and editing; Anne Gresch, Investigation, Visualization, Writing – review and editing; Isabella Marinelli, Investigation, Methodology, Writing – review and editing; JaeAnn M Dwulet, Software, Methodology; David J Albers, Writing – review and editing; Vira Kravets, Data curation, Formal analysis, Methodology, Writing – review and editing; Richard KP Benninger, Conceptualization, Supervision, Funding acquisition, Investigation, Methodology, Writing – original draft, Project administration, Writing – review and editing

### Author ORCIDs

Jennifer K Briggs https://orcid.org/0000-0002-8737-2215
JaeAnn M Dwulet http://orcid.org/0000-0003-2519-5193
Vira Kravets http://orcid.org/0000-0002-5147-309X
Richard KP Benninger http://orcid.org/0000-0002-5063-6096

## Ethics

All animal procedures were performed in accordance with guidelines established by the Institutional Animal Care and Use Committee of the University of Colorado Anschutz Medical campus (protocol 000024). All surgeries were performed under ketamine/xylazine anesthesia, with minimal discomfort to the animals.

## Decision letter and Author response

Decision letter https://doi.org/10.7554/eLife.83147.sa1
Author response https://doi.org/10.7554/eLife.83147.sa2

---

## Additional files

### Supplementary files
• MDAR checklist

### Data availability

Raw microscopy imaging data is available on the EMBL-EBI-supported BioImage Archive. Analysis code, model code, and simulated data is available via GitHub at https://github.com/jenniferkbriggs/Functional_and_Structural_Networks (copy archived at *Briggs, 2023*).

The following dataset was generated:

| Author(s) | Year | Dataset title | Dataset URL | Database and Identifier |
|---|---|---|---|---|
| Benninger R | 2024 | Beta-cell intrinsic dynamics rather than gap junction structure dictates subpopulations in the islet functional network | https://www.ebi.ac.uk/biostudies/bioimages/studies/S-BIAD1024 | EBI BioImage Archive, S-BIAD1024 |

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
