## [Editor Report]

The paper uses both computational and laboratory approaches to test the hypothesis that connectivity in β cells within the islet is due to metabolic rather than gap junctional coupling efficacy. This will be an important advance for understanding the role of heterogeneous β cell populations in driving synchronized oscillations by islets and by extension the oscillatory insulin secretion observed in vivo. There will be implications of the work for understanding the mechanisms of type 2 diabetics and β cell function in general.

---

## [Decision Letter]

**Decision letter after peer review:**

Thank you for submitting your article "Β-cell Metabolic Activity Rather than Gap Junction Structure Dictates Subpopulations in the Islet Functional Network" for consideration by *eLife*. Your article has been reviewed by 5 peer reviewers, and the evaluation has been overseen by a Reviewing Editor and Naama Barkai as the Senior Editor. The following individuals involved in the review of your submission have agreed to reveal their identity: David J Hodson (Reviewer #1); Andraz Stozer (Reviewer #2); Victoria Salem (Reviewer #3).

Essential revisions:

1) The modeling, based on the Cha et al. model is clearly of fast oscillations but the data shown include both fast and slow oscillations. The model will therefore need to be revised to simulate the slow data more closely either using the Cha model in a different mode, or a different model.

2) There are questions about some of the analyses and plots only showing a small number of data points, which must be addressed and also the choice of R-value as discussed.

3) There was a general concern about the small-worldness of the data and this must be dealt with.

4) It is recommended that some of the key conclusions made by the authors be more tempered as some of the reviewers commented that perhaps some key conclusions were overstated based on the data and analysis.

5). It would help if the authors more clearly indicated which findings were truly novel and which were confirmation of observations made previously by their lab and others.

6). The manuscript is quite long and rather dense and thorough editing must be done to improve its readability (especially by non-experts); this would help the paper.

*Reviewer #1 (Recommendations for the authors):*

I have the following constructive suggestions to further improve the studies:

– "In diabetes the islet shows disrupted [Ca^2+^] synchronization (representing a disrupted functional network) and diminished gap junction coupling (representing a disrupted structural network)." A schematic would be helpful here to visually explain the functional and structural networks that are the focus of the studies.

– The resolution of the FRAP studies is impressive, with single-cell definition. Nonetheless, FRAP only looks at cationic dye transfer, does not report conductance, and is unlikely to be as sensitive as patch clamp measures. Conversely, patch clamp is inherently biased, is spatially limited, and likely selects for certain cell subpopulations (i.e. cells that do not form seals or respond to voltage ramps are disposed of). Caveats are needed here, since there is probably no perfect measure of GJ conductance across a cell population, which is the basis for the modelling here.

– "Prior work suggested that cells with highly synchronized Ca^2+^ oscillations possess both elevated metabolic activity (in agreement with our observations, Figure 2) and high levels of gap junction conductance (unlike what we observe experimentally, Figure 2)". Johnston et al. showed that Gjd2 knockdown disrupted synchronization, functional connectivity, and hub number, implying a role of GJ communication in hub function. However, this knockdown was not hub-specific so any inferences about hub GJ coupling are indiirect.

– "However, as the functional network becomes sparser, the uncertainty in the random network-based approach increases (Supp. 4d, seen more clearly in Supp. 6d)." Maybe I am not following here, but would a sparser functional network not be expected to decrease power-law slope/fit and hence the definition of small-worldness? Some more explanation would be helpful.

– "Semi-quantitative immunofluorescence previously demonstrated elevated glucokinase protein in β-cell hubs, suggesting elevated metabolic activity." The authors also looked at TMRE accumulation in hubs, showing more hyper-polarized mitochondria indicative of increased OXPHOS (hence fitting with the NAD(P)H).

– It is well established that gap junction coupling is decreased in animal models of diabetes, as well as aged/high BMI human islets. However, are functional and structural networks mutually exclusive? That is, do changes in cell states drive changes in GJ coupling or vice versa? Or is there no relationship between these parameters?

– Electrical coupling is assumed to be a static phenomenon within the islet. However, connexins are dynamic proteins, their gating can be influenced by cAMP/PKC signals (shown by this group), and expression levels differ across β cells. How would the authors predict such heterogeneity to influence their model? How might this be built into analyses going forward?

– A comment on the translation of the results to human islets would be appreciated since there are known differences in coordination (more regional) and cell interactions compared to rodents.

– "Finally, the islet also contains α-cells, δ-cells, and other cell types which can influence β-cell dynamics via paracrine mechanisms". It would be worth citing recent studies from the Chen and Tang groups (DOI s41467-022-31373-6).

– The authors note some study limitations. It would be worthwhile also discuss limitations with the models of GJ disruption. For example, GJD2 KO is global and occurs early in development, so any effects may be confounded by β cell de-differentiation/immaturity (and metabolic changes therein). I understand that conditional Cx36 models have been historically unavailable, but conditional-ready KOMP mice could be informative in the future.

*Reviewer #2 (Recommendations for the authors):*

Introduction

– I suggest a reference for the epidemiological data, i.e., for the number of people affected by diabetes. The IDF Diabetes Atlas is a valuable resource and typically accompanied by citable publications, such as PMID: 34879977

The first part of the Results: Cellular metabolism, but not elevated gap junction coupling, is observed in highly synchronized cells in a simulated β-cell network, the corresponding Figures, and Methods

– Rth was chosen at the value of *R*th = 0.9995 – this is a rather high value compared with values typically chosen for experimental calcium traces (e.g., Stožer PLoS Comput Biol 2013, Front Endocrinol 2022), which indicates that the model did not produce large temporal differences/delays between cells. Could this be improved in the model to more closely mimic the experimental situation where delays between cells are on the order of magnitude that is the same as the order of magnitude of burst/fast oscillation durations? If there are objective reasons that this cannot be done easily with existing models (by for instance destabilizing the model with increasing heterogeneity), I suggest that the authors point out this difference between experiments and model and more explicitly address the nature of this discrepancy.

– From the inset in Figure 1d, it is also not clearly visible what the temporal delay between traces in different cells was. Please, provide a more detailed inset/zoom-in.

– In addition to the mean parameter values, I recommend that the authors also provide the range of values more precisely quantify heterogeneity. Since the statistics in Figure 1 are based on a rather small number of simulated islets (N=5), I suggest that effect sizes be reported as well.

– In the model, only 500 seconds were simulated, and then the so-called fast calcium oscillations present during the second (plateau) phase of the calcium response (and brought about by bursts of electrical activity) were analyzed. In the simulated traces, they seem to be around 15-25 seconds long (Figure 1d). From the Methods section, it is not entirely clear what period of calcium traces obtained by calcium imaging in isolated islets was used for the network analysis. The authors state that "[Ca^2+^] time courses were analyzed during the second-phase [Ca^2+^] response when the slow calcium wave was established", but Figure 2 and other parts of the manuscript do not provide enough information to be sure whether the interval used for network analyses included the whole traces beyond approximately 500 seconds (Figure 2b) or only smaller parts of these traces that would correspond in duration to intervals used on simulated traces. I think it is critical that the authors address and resolve this question in detail for the following reasons.

– First, simulated traces reflect only fast oscillations, whereas in experimental traces (See Figure 2b), there are a few slow oscillations that last approximately 300 seconds each, and superimposed on them fast oscillations (approximately 10 seconds long), corresponding to oscillations analyzed on simulated traces. Since there may be fundamental mechanistic differences between fast and slow oscillations, the first being more importantly determined by ion channel properties and electrical coupling and the second more importantly by metabolism, the networks constructed from experimental traces may contain and provide information that is different from the information provided by simulated traces and networks based on them. If the authors included in their analyses on experimental traces only one part of a slow wave (e.g. 200-300 seconds) containing a few fast oscillations (e.g. 10 fast oscillations), then the simulated traces and networks can be compared with experimental traces and networks. If not, i.e., if they included longer periods (e.g. 1000 seconds) containing a few slow oscillations and thus many more fast oscillations (e.g., 50 fast oscillations), then the Pearson correlation between traces may heavily depend on the slow trends defined by slow oscillations (as addressed recently in Zmazek et al. Front Physiol 2021 and Stozer et al. Front Endocrinol 2022), and thus the network metrics convey entangled information on both fast and slow properties. Most importantly, the main conclusion from the experimental part that the rate of metabolism may be more important than other factors in determining functional network properties may be due to the prevailing influence of metabolic oscillations on networks based on experimental recordings. In this case, I suggest two possible solutions. The authors could extend the simulations to produce a mixed pattern of oscillations (which should be possible, given the existing work by Bertram, Sherman, Satin, Nunemaker, and Benninger) and then systematically analyze the impact of the fast and slow components on network properties and the importance of different parameters (kglyc, gKATP, gcoup) and compare the findings with experimental data where they could also separately study the two components. Alternatively, the authors could extract the fast component from the experimental traces (by using an appropriate filter) and limit their analyses and discussion to the fast oscillations only.

– Second, resolving the above point is not only important for this manuscript, but also for resolving the (apparent) differences between studies (and the Rutter-Rorsman dispute).

– Finally, resolving the first above point is also particularly important for extending the findings to Cx36-/- mice where fast and slow oscillations may be affected to a different extent by the absence of gap junctions, with desynchronized fast oscillations and possibly less desynchronized slow oscillations.

The other parts of the manuscript

– My comments and suggestions to other parts of the manuscript depend on how the above fast-slow oscillation issue will be resolved since all of the following Results and corresponding Figures strongly depend on the way how networks are constructed. I will be more than happy to do so in a possible next round of revision.

*Reviewer #3 (Recommendations for the authors):*

Briggs et al. present a strong set of both computational and experimental approaches to investigate some current controversies in the functional relevance of β cell heterogeneity and pancreatic islet function. They conclude that β cell connectivity is driven by metabolic rather than gap-function-mediated structural coupling. This lays the ground for future studies understanding how metabolic coupling relates to the identity of "hub cells" and to what extent this can be targeted in the treatment of diabetes. Overall this is a very strong paper that is of interest to the readership of a journal like *eLife*. It is extremely well written. I would suggest that more experimental data for the NADPH and FRAP experiments might build up confidence in the finding that connectivity is defined by metabolic coupling – at present, the datasets are convincing that metabolic coupling is absent in cells that lack calcium connectivity but are not convincing enough in the opposite direction.

Here are my general comments.

The authors state that "Functional networks represent the emergent system behavior". Diabetes ensues from the autoimmune destruction of β cells (T1) or the functional demise (a mix of environmental insult with a genetic predisposition – T2). To what extent can network theory really tell us anything about the pathophysiology of these disease states or aid the development of new treatments? Or is this simply another tool, like GSIS, for assessing islet dysfunction? I think it would be useful to have this sort of oversight mentioned in the discussion, to persuade readers of the real relevance of your work.

"cell hubs can exert a strong influence over islet dynamics21,24,25 53 and are preferentially disrupted in diabetes23" Did the Johnston paper really conclude that diabetes is causally related to the preferential loss of hubs over followers? Or is it more likely that in a dysregulated islet loss of "hubs" becomes more apparent? I personally would have prefaced this paper by summarising the mathematical and experimental evidence for hubs that exhibit small-world properties.

When you say "structural location" what do you mean – you didn't for example look at a position relative to blood vessels, α cells, nerve endings, and δ cell projections. As a general comment, I'm not really comfortable with you equating conductance (ie the physical number of gap junction interfaces between β cells) as synonymous with their structural topography. There are so many unknowns here still – for example the contribution of other endocrine cells, nerves, and "humoral factors". Your computational model essentially randomly assigns heterogeneity in terms of metabolic rates, for example, across the 1000 "β" cells. In fact, this may not reflect the fact that such heterogeneity is indeed based on a cell's position relative to other aspects of the islet microenvironment (not just other β cells).

Results section 1

Briggs et al. start with data extracted from a model. Whilst the thresholds selected are generally well justified and the testing of various thresholds adds robustness, it is, in the end, just a model. It has been well validated by electrophysiologists, but the audience of this paper is wider and I think it is worthwhile reminding us in a couple of sentences of what the major inputs are into this model and how that results in oscillatory behaviour (I know you do this in supplemental anyway). Presumably, this is being run for a certain ambient glucose level ie parameters change between low and high glucose. It's important that you point out there is no accounting for other endocrine cells etc – which I think you do in the

Discussion.

Was such a high R-value necessary to get the requisite power-law type distribution a surprise to you? Were all 1000 cells included in your readouts? Why did you choose the 60% connection cut-off to define hubs? Is this done by others? In your supplementary data as you move (in very tiny increments) across R thresholds the number of connections on your axis changes apparently randomly (the scale goes from 0-200 to 0-800). Why?

The differences in Kglyc between hubs and followers are statistically significant but the difference in absolute terms seems very small – is this relevant?

Results section 2

This is the experimental dataset which is very interesting.

"We extracted the functional network (Figure 2c) and again generated a normalized degree distribution which reflects a scale free-like distribution."

– ok but there is no mention of actual R values here – presumably much less than the modelled ones previously and the cut off for defining hubs on normalised degree of "edginess" is also different from the above (and others?). Conversely, you then introduce the concept of a "low degree" cell without defining that.

Just out of interest, the cross-section shows that some β cells are much more fluorescent than others, presumably reflecting variation in GcAMP expression. Do you think this is an issue for your calcium trace analysis?

In the Methods section please can you explain in a bit more detail how you extracted your NADPH data – over what time period/resolution etc.

"Furthermore, the NAD(P)H response trended lower in cells that were functionally disconnected (Ca^2+^ 128 oscillations lacking any synchronization), compared to connected cells (Figure 2g)."

Presumably, you only looked at calcium oscillating β cells as inactive ones will obviously likely have no other How did you define an oscillation? Your time course of over 30 minutes looks long. The association between low NADPH signal and low connectivity seems much more robust than that between hubs and high NADPH signal. Would repeat experiments firm this up?

Is the FRAP experiment powered? I don't have a feel for the sensitivity of this method to pick apart quantifiable differences in gap junction connections but the numbers here seem low – only 4 islets.

Results section 3

The next section poses the question what does the islet functional network indicate about its underlying structure or intrinsic properties on an individual cell basis?

The authors appear to have returned to their simulated data here which initially confused me so should be headlined at the outset. Given that the EPists know that GJ coupling cannot explain connectivity across more than a few cells, I think it's important to state in the main text that you enforced spatial limits on your structural connectivity analyses.

Some of the surprising findings e.g.

"The probability that two cells were synchronized in the functional network, given that they shared a gap junction in the structural network, was = 0.39" has been discussed well later (eg where they don't tally with prior experimental measures.

On the whole, I find the ending statement "These results further indicate that metabolic activity, not gap junction connections, is a greater driving factor for cells to show high [Ca^2+^] synchronization and thus influence the functional network." to be robust.

The next section looks at long-range functional connections which traverse cells, quite dense. I thought the experiment that modelled GJs of higher conductance to be rather extraneous and could have gone into supplemental but I don't feel strongly about this.

I think there needs to be some unpacking of the term kglyc as a measure of "glucose metabolism" – how does the sameness of kglyc translate into closer coordination of calcium oscillations? Is this simply the speed with which glucose sensing and insulin release are being cycled? Is metabolic "coupling" simply a coincidence – is this all simply an epiphenomenon?

Finally, the authors look at experimental data from connexin KOs (het and homs vs WTs). I found this data (even though we have seen this before) to be rather surprising given the previous narrative. In essence, what it shows is that a 50% reduction in GJs has a marked effect, and a total loss results in a near-complete loss of hubs. If connections, and therefore presumably networks that deliver hubs, can exist just based on random metabolic homogeneity in distant β cells alone, why does this happen? Is there any NADPH data from the connexin KO islets? With the loss of gap junctions, I can see why the amplitude of the calcium oscillations in β cells might become smaller, but why do other metrics of the oscillations eg frequency seem to change?

Figures

Figure 1 – Kglyc and g coup need to be defined/explained so the figure stands alone.

Figure 1b – the word seed is confusing/comes out of the blue. Do you actually mean "modelled islet"?

Figure 1 f/g – this may display my incomplete understanding of the different populations you defined and measured, but why was a paired t-test used to compare them and if these readings are truly paired were they really normally distributed?

Figure 2 – the defined cellular ROIs in a is not the same as the ones shown on your map in c – I was expecting them to be!

Again, here we have the notion of Kmax mentioned which needs defining in the legend.

*Reviewer #4 (Recommendations for the authors):*

1. The paper is very dense and complex, and thus is difficult to follow, and as a consequence will only be understandable to a very specific and likely small group of readers. The methods, the analysis, and the models used, while largely based on prior work require careful reading, and a true understanding of their implications will be mostly lost on most readers. The paper would therefore be improved by reducing its size and complexity and removing excessive verbiage. Some of the more mathematical aspects could be relegated to either previously published papers by the group or placed in the Supplementary Material. It might be worth considering breaking the paper into two different but complementary ones, one emphasizing theory and the other measurements but I realize this would make reading the two simultaneously very difficult.

2. The paper would also be improved by very clearly highlighting what is truly new in the paper and deemphasizing what is a restatement of what has been known already (e.g. discussing how the islet can be considered a small world network, etc).

*Reviewer #5 (Recommendations for the authors):*

I have a major concern about how cells related by cell metabolism and cells related by gap junctional coupling are treated as independent. There is no mechanism mentioned (?) for metabolism to synchronize islet cells independent of gap junctions. Consequently, I have concerns about the conclusions of the results.

This work seems to suggest a disregard of the necessity of gap junctional connections for islet synchronization, and suggest that functional connectivity – based on statistical correlation of traces – somehow predominate…up until Figure 6 where knocking down gap junctions is acknowledged as important.

I think identifying subpopulations is reasonable, but when a 60% functional connection threshold is what defines "hub" cells and that produces between 50 and 200 cells, "hub" seems to lose its meaning, especially when these "hubs" seems to be all grouped together rather than dispersed as might be expected in a small world network. The small network-ness is questionable. The sensitivity to the correlation threshold suggests a certain lack of robustness. Perhaps I am misunderstanding the number of cells labeled as "hub cells."

The excitability of cells via metabolism, for example, must be communicated through some (often structural) mechanisms. I appreciate that this article is attempting to get at how that division breaks down and I think much of the calculation and simulation is useful, but the language expressing the certainty rather than the more appropriate equivocation is not completely appropriate.

[Editors’ note: further revisions were suggested prior to acceptance, as described below.]

Thank you for resubmitting your work entitled "Β-cell Intrinsic Dynamics Rather than Gap Junction Structure Dictates Subpopulations in the Islet Functional Network" for further consideration by *eLife*.

The manuscript has been improved but there are some remaining issues that need to be addressed, as outlined below:

*Reviewer #2 (Recommendations for the authors):*

I much appreciate the effort that the authors put into addressing my main skepticism about the paper, namely the fact that one type of oscillations were analyzed for experimental data and another type for modelled data, by including an additional model for slow oscillations and additional experimental data.

Given the new results, below, I provide my additional comments and suggestions that should not be too difficult for authors to consider and take into account.

110-197 and Figures 1-S3:

First, the difference between hubs and non-hubs in terms of kglyc is around 0.14 vs 0.13, whereas for gcoup it is around 0.75 vs 0.6. In relative terms, the former is an 8 % difference and the latter a 25 % difference. It is true that the former is more significant statistically, but this is due to the fact that the heterogeneity is larger in the distribution of gcoup for hubs! Therefore, to me, it seems that both kglyc and gcoup seem to define the hub cells to a comparable degree. This conclusion seems to also be more compatible with findings for slow oscillations where the relative difference between hubs and non-hubs for gcoup is again around 20-30 %.

I suggest that authors consider this under the corresponding sections of results and discussion, and perhaps to some extent also in Figure 8.

Second, at a threshold for hub cells of a 60 % normalized degree (i.e., the criterion for the separation line between hubs and non-hubs), different percentages of β cells qualify as hubs for different simulated islets, as evident in Figure 1b. More specifically, the cumulative number of cells qualifying as hubs for islets 3 and 5 is much more than the cumulative number of hubs in simulated islets 1 and 2. This differs from the criterion used in other important recent studies, i.e., in Johnston et al. Cell Metab 2016, hubs typically represent 1-10 % of islet cells, in Lei et al. Islets 2018, hubs represent 10 % of cells, and in Stozer et al. Am J Physiol 2021 and in Sterk et al. Biophys J 2023, hubs represent 1/6th of islet cells. I think it would enhance the comparison between the present and previous research if the authors provided the % of hub cells per islet. Additionally, setting the line of separation between hubs and non-hubs at the given threshold (not changing the threshold and thus the distribution of normalized degree) individually for each islet to achieve a fixed percentage of hub cells per islet close to 10 % of the most connected cells, would be a valuable addition to supplemental figure 1 and could perhaps help detect a larger (or smaller) difference between this more extreme group of cells and the majority of non-hub cells, in terms of kglyc, gkatp, and gcoup. Performing this additional analysis at least for the threshold value used in the main Figure 1 could add much value to the manuscript and make the findings even more directly comparable with the aforementioned studies. The same suggestion could also be taken into account for the analyses of modelled slow oscillations and for the experimental analyses of metabolic activity and coupling, where 10 % of the most connected cells could be considered/classified as hubs as well. Such additional analysis should be easily feasible since it considers a fraction of cells that have already been used in the analyses, but their degree of "hubness" is probably more.

Given that the duty cycle (percentage of active time) so strongly correlates with the role of hub cells, it is somewhat surprising that the authors do not mention in the discussion that this same finding has recently been obtained for both mouse and human islets (Šterk et al. Biophys J 2023, Stožer et al. Am J Physiol 2021, Gosak et al. Diabetes 2022).

Third, I do value the effort of authors in replying to my comment regarding the exceedingly high correlation between simulated traces compared with experimental traces. However, besides noise, one very important aspect is the choice of model and the parameters which determine the observed lags between different cells or the wave speed. For instance, Cappon and Pedersen in their Chaos article built on the model used in Benninger Biophys J 2008 to produce lags between cells that are very realistic compared with experimentally observed values, whereas lags in the present study are just a fraction of the duration of a burst (fast Ca oscillation).

At present, I can only speculate about that, but the rather low conditional probability that two cells sharing gap-junctional conditions also show a functional connection could also be a consequence of the exceedingly short time lags between signals in the model employed in this study. More specifically, in the model by Cappon and Pedersen, there are waves travelling across islets and direct neighbors have shorter phase-lags between their signals compared with more distant cells and therefore on average more similar signals, similar to experimental recordings in isolated islets and tissue slices (the velocity being around 100 um/s, the time lag between direct neighbors is around 0.1 s and the time lag between most distant cells is around 1-2 s). This means that the values of R will be higher for direct neighbors and at any given threshold they will be functionally connected with a higher probability than with more distant cells where lags can be an order of magnitude more.

Therefore, I would suggest that in the future (not in this study), authors also repeat some of the analyses with modeled traces that show larger lags. This would probably also enable them to explore the relationship between different modelled/analyzed parameters and the role of pacemakers, i.e., wave initiators and other populations of cells. Perhaps, if authors consider the above suggestion as useful, they could include it in the discussion as a possible drawback of the present model and as a suggestion for future studies.

*Reviewer #5 (Recommendations for the authors):*

The authors have thoroughly responded to my (and others') reviews. I think their modifications are reasonable and have tempered the language appropriately. The results are somewhat interesting and the techniques employed to measure and analyze the islets including adding an additional model system are extensive. They have addressed my points.

One point: We discussed (Reviewer #5 question/response 5b) the potential outlier at 2.35 of Figure 1f (gkatp data – Β Cell Hubs). That data point no longer exists on the graph (that I can see?). Since they argue convincingly that point should in fact not be treated as an outlier, I expect it is an oversight of it not being there. This should be fixed (or explained) prior to moving forward with publication.

---

## [Author Response]

Essential revisions:1) The modeling, based on the Cha et al. model is clearly of fast oscillations but the data shown include both fast and slow oscillations. The model will therefore need to be revised to simulate the slow data more closely either using the Cha model in a different mode, or a different model.

In addition to the Cha et al. based model, we created a coupled simulation of the Integrated Oscillator Model (IOM) which was derived to simulate both fast and slow oscillations. Results from this coupled IOM are described in Figure 2, and support our overall conclusions. These essential revisions are included in Figure 2, Results page 6 lines 128-156.

We also conducted additional experimental measurements of calcium imaging and fluorescence recovery after photobleaching (FRAP) for which we separately analyzed slow, fast, and mixed [Ca^2+^] oscillations. For all three oscillation types, we found no relationship between functional network degree and gap junction permeability. (Figure 3i,j and Figure 3 Supplemental 1c-h). These essential revisions are included in Figure 3i,j, Figure 3 Supplemental 1c-h, and Results page 7 lines 184-186 and lines 188-191.

2) There are questions about some of the analyses and plots only showing a small number of data points, which must be addressed and also the choice of R-value as discussed.

We have clarified what each data point represents in both the captions and the results text, as well as in the response to reviewers below. Briefly, in simulations, each dot corresponds to the averaged value in a simulated islet (Figures 1, 2, 4, 5, 7). We conducted five independent simulations for both the Cha-Noma and IOM model. In experimental results, each data point also corresponds to the average value over an islet (Figures 3, 6). These essential revisions are included in all figure captions.

Importantly we have conducted additional experiments to address any concerns about the small number of experimental data points. These essential revisions are included in Figure 3j,k.

We have responded to the reviewers regarding the choice of threshold used in the simulations and experiments. The simulation threshold required to match past experimental results is high, primarily because the simulations lack stochastic noise, which strongly influences the correlation threshold. There are also additional computational and model limitations with respect to the Integrated Oscillator Model which reduce the amount of heterogeneity in the simulated [Ca^2+^] oscillations. We describe these limitations in our discussion and include three supplemental figures showing the effect of threshold on simulation (Figure 1 Supp 1, Figure 2 Supp 1) and experimental (Figure 3 Supp 1) results. With the exception of the coupled Integrated Oscillator model (which we further discuss), the threshold does not influence our conclusions. These essential revisions are included in Supplemental Figures 1-3, Discussion page 14 lines 448-455.

3) There was a general concern about the small-worldness of the data and this must be dealt with.

We agree that we did not conduct a comprehensive small-worldness analysis of our data. This was not a primary focus of the manuscript as small-worldness of the functional network has been shown in Stozer 2013[1]. In efforts to address essential revisions 5 and 6 and simplify the manuscript, we removed all small world analyses from the previous figures 5, 6 and associated supplemental figures. These essential revisions are addressed by the removal of figures and associated text.

4) It is recommended that some of the key conclusions made by the authors be more tempered as some of the reviewers commented that perhaps some key conclusions were overstated based on the data and analysis.

We agree that our initial manuscript miscommunicated some of our interpretations of the results. We have tempered key conclusions and statements throughout the manuscript. One important miscommunication in our initial manuscript is that we are not claiming that metabolic coupling drives the functional network instead of electrical coupling. Rather, intrinsic properties of β-cells drive variations in the functional network rather than structural gap junctions. Gap junctions are still critical to synchronize oscillations across the whole islet. We have taken extensive efforts to remove this miscommunication from the manuscript. This includes a sentence stating this in the discussion (Page 12, Lines 394-397), changing the title to “Β-cell Intrinsic Dynamics Rather than Gap Junction Structure Dictates Subpopulations in the Islet Functional Network”, and altering writing throughout the manuscript to be clearer (particularly in the Results and Discussion). We also highlight data that show the importance of gap junctions so we do not incorrectly communicate that structural coupling is unnecessary for islet synchronization. These essential revisions are addressed in the Introduction (page 3 lines 45-47) by a change in the title, by highlighting data in Figure 6 and 7 and in the Results (page 7 lines 203-205, page 8, lines 241-242, page 9 lines 256 and 293-294, page 10 lines 308-309) and Discussion (page 11 line 318-319, page 13, lines 434-435).

5). It would help if the authors more clearly indicated which findings were truly novel and which were confirmation of observations made previously by their lab and others.

We have substantially rewritten the introduction and Discussion sections to address this comment.

6). The manuscript is quite long and rather dense and thorough editing must be done to improve its readability (especially by non-experts); this would help the paper.

In response to comments 5 and 6, we have significantly restructured the paper –the Introduction, Results and Discussion – with careful emphasis on highlighting which of our findings are novel and which have been shown in previous papers. We also were intentional about making the manuscript less dense and more accessible. To be specific, we reduced question 3 by removing any network metrics that we felt were tangential to the goals of the study and removing additional peripheral data. This includes comments about small world-ness and local efficiency and simulated data correlating metabolism and gap junction coupling. Despite including an additional figure, other additional data, and additional discussion to address comments by all 5 reviewers, the word count remains similar. These essential revisions are addressed throughout the Introduction, Results and Discussion, in Figures 6 and 7, and through removing Supplemental Figures.

Reviewer #1 (Recommendations for the authors):I have the following constructive suggestions to further improve the studies:– "In diabetes the islet shows disrupted [Ca^2+^] synchronization (representing a disrupted functional network) and diminished gap junction coupling (representing a disrupted structural network)." A schematic would be helpful here to visually explain the functional and structural networks that are the focus of the studies.

We thank the reviewers for a useful suggestion on how to make the paper clearer. We have added an inset into Figure 8 that shows the structural and functional networks.

– The resolution of the FRAP studies is impressive, with single-cell definition. Nonetheless, FRAP only looks at cationic dye transfer, does not report conductance, and is unlikely to be as sensitive as patch clamp measures. Conversely, patch clamp is inherently biased, is spatially limited, and likely selects for certain cell subpopulations (i.e. cells that do not form seals or respond to voltage ramps are disposed of). Caveats are needed here, since there is probably no perfect measure of GJ conductance across a cell population, which is the basis for the modelling here.

We thank the reviewer for their enthusiasm for FRAP measurements, which has single cell resolution as we previously demonstrated by Farnsworth et al. [2]. We agree FRAP will not be as sensitive as the gold standard patch clamp measurements, but it does allow multiple cells to be assessed and compared in the same islet. We do note Cx36 is cationic specific so is suited to FRAP here (FRAP of anionic dyes in the islet reveals minimal dye transfer). We now mention caveats associated with FRAP within the discussion. See Discussion page 14 lines 457-464.

– "Prior work suggested that cells with highly synchronized Ca^2+^ oscillations possess both elevated metabolic activity (in agreement with our observations, Figure 2) and high levels of gap junction conductance (unlike what we observe experimentally, Figure 2)". Johnston et al. showed that Gjd2 knockdown disrupted synchronization, functional connectivity, and hub number, implying a role of GJ communication in hub function. However, this knockdown was not hub-specific so any inferences about hub GJ coupling are indirect.

Thank you for pointing out this need for clarification. The prior work were referring to was in Lei et al. 2017 [3] who conducted a computational study of the functional network in human islets and concluded that hub cells must have stronger than average gap junction conductance in order to silence the islet, as first demonstrated in Johnston et al[4]. We therefore repeated our analysis on simulated islets whose gap junction conductance was proportional to metabolic activity. However, we note that in seeking to make the manuscript shorter and easier to read we have removed this Figure and associated Results text. Nevertheless, when examining Cx36+/- islets we do see a reduction in the network degree consistent with the results the reviewer refers to. However as suggested this is likely due to genetic knockout not being hub specific. We now mention this – See Discussion page 12 lines 372-374.

– "However, as the functional network becomes sparser, the uncertainty in the random network-based approach increases (Supp. 4d, seen more clearly in Supp. 6d)." Maybe I am not following here, but would a sparser functional network not be expected to decrease power-law slope/fit and hence the definition of small-worldness? Some more explanation would be helpful.

We thank the reviewer for this thoughtful comment. Indeed, in a sparse network the power-law fit will decrease exactly as the reviewer says. In the comment above, we refer to the increase in the uncertainty of small worldness in the random networks. In order to establish whether the functional network topology comes from interesting natural behavior or is a result of randomness, we compared the network metrics (such a “small-worldness”) to random graphs. The random graphs are akin to a null hypothesis. This technique is well established in network theory. The random graphs are formed by taking the same number of edges and nodes as in the network of interest and randomly exchanging the edges with some probability. This was described in *Table 1* and *Methods Network Topology Analysis: Random Networks***.** The comment above refers to the analysis of these random graphs. As the functional network becomes sparser, the configuration of edges becomes more variable and small changes in the random graph have a larger impact on the small-worldness parameter. Therefore, the uncertainty in this small-worldness increases.

However, we note again that in seeking to make the manuscript shorter and easier to read we have removed results relating to small worldness, and rather present the clustering and global efficiency data (Figure 6, 7).

– "Semi-quantitative immunofluorescence previously demonstrated elevated glucokinase protein in β-cell hubs, suggesting elevated metabolic activity." The authors also looked at TMRE accumulation in hubs, showing more hyper-polarized mitochondria indicative of increased OXPHOS (hence fitting with the NAD(P)H).

Thank you for pointing out these additional measurements- we now refer to the TMRE measurements which are consistent with the NAD(P)H measurements. See Discussion page 12 lines 368-370.

– It is well established that gap junction coupling is decreased in animal models of diabetes, as well as aged/high BMI human islets. However, are functional and structural networks mutually exclusive? That is, do changes in cell states drive changes in GJ coupling or vice versa? Or is there no relationship between these parameters?

We thank the reviewer for this question. In Figure 5 and 6 (original version, now figure 6 and 7), we analyzed the effect of changes in structural networks on functional networks. As the reviewer indicates, in diabetes and other pathological conditions, gap junction coupling is reduced. Therefore, we analyzed how the functional network changes as gap junction coupling (e.g., the structural network is decreased). Our data suggest that functional and structural networks are not mutually exclusive. Decreasing edges in the structural network do lead to decreased oscillation synchronization and, therefore, average degree in the functional network. We also found that the topology of the functional network is affected by the structural network. Data in figure 5 and Figures 6-7 indicate that gap junction coupling is influencing longer range synchronization across the islet, which explains the minimal overlap between functional and structural network on a node or edge scale, but that the networks are not mutually exclusive. We now describe this further. See Discussion page 13 lines 426-427.

– Electrical coupling is assumed to be a static phenomenon within the islet. However, connexins are dynamic proteins, their gating can be influenced by cAMP/PKC signals (shown by this group), and expression levels differ across β cells. How would the authors predict such heterogeneity to influence their model? How might this be built into analyses going forward?

The reviewer is correct and raises an important point that connexin trafficking is dynamic and Cx36 gap junctions can gate, in response to cAMP signals; and also show expression changes in response to Ca^2+^ – CaN – NFAT signaling. Further, Cx36 gap junction permeability varies between cells (as we include in the model). This variation would be expected to be on the time scale of minutes (cAMP-gating) to hours (cAMP-trafficking) to hours-days (NFAT-transcription) which is much slower than electrical changes that we analyze. Further Cx36 gap junctions are only weekly voltage gated and therefore we can consider gap junction conductance to be static in relation to electrical oscillations.

Against our initial prediction, the synchronization between cells is not substantially influenced by variations in the local gap junction conductance (Question 1). As such we would conclude that if the strength of gap junction coupling varies between cells over time, providing the islet-average conductance remains unchanged, the positioning of hub cells should not change significantly. Thus, if positioning of hub cells does change over time, this would be predicted to result from changes in cell-intrinsic properties. We briefly discuss this – See Discussion page 14 lines 465-468.

Variations in gap junction conductance do seem to influence another cell sub-population – first responder cells (see Kravets et al. PLOS Biology 2022[5]) and the ability of a cell to transition between quiescence and activity upon nutrient stimulation. However, while interesting we avoid further complicating the manuscript by introducing this additional topic.

– A comment on the translation of the results to human islets would be appreciated since there are known differences in coordination (more regional) and cell interactions compared to rodents.

Thank you for this note. We have added the following sentence to the discussion:

“This [referring to future analysis in the influence of other cell types] is particularly important when translating our findings to human islets because α-cells, δ-cells, and β-cells are known to be more mixed in human islets”*.* See Discussion page 14 lines 475-477.

– "Finally, the islet also contains α-cells, δ-cells, and other cell types which can influence β-cell dynamics via paracrine mechanisms". It would be worth citing recent studies from the Chen and Tang groups (DOI s41467-022-31373-6).

Thank you for the suggestion. We agree that the Chen and Tang paper is very intriguing. We added the citation and comment: “Additionally, β-cell oscillation types (slow, fast, or mixed) may be directly related to the number of α-cells present in the islet[6]. ” See Discussion page 14 lines 472-473.

– The authors note some study limitations. It would be worthwhile also discuss limitations with the models of GJ disruption. For example, GJD2 KO is global and occurs early in development, so any effects may be confounded by β cell de-differentiation/immaturity (and metabolic changes therein). I understand that conditional Cx36 models have been historically unavailable, but conditional-ready KOMP mice could be informative in the future.

We do appreciate the Cx36 ko model has caveats being a global knockout. While changes in β cell specification in such models have not been reported we understand this cannot be excluded. Similarly, while unperturbed floxed models have not been available, these should be available in the future. However, we do note that dissociated β cells from both wild-type and Cx36ko islets show similar insulin secretion responses. In response to this comment we mention potential caveats associated with the Cx36 global ko model. See Discussion page 14 lines 459-464.

Reviewer #2 (Recommendations for the authors):Introduction– I suggest a reference for the epidemiological data, i.e., for the number of people affected by diabetes. The IDF Diabetes Atlas is a valuable resource and typically accompanied by citable publications, such as PMID: 34879977

Thank you for the reference, this has been added to the paper. See Introduction page 3 lines 42.

The first part of the Results: Cellular metabolism, but not elevated gap junction coupling, is observed in highly synchronized cells in a simulated β-cell network, the corresponding Figures, and Methods– Rth was chosen at the value of Rth = 0.9995 – this is a rather high value compared with values typically chosen for experimental calcium traces (e.g., Stožer PLoS Comput Biol 2013, Front Endocrinol 2022), which indicates that the model did not produce large temporal differences/delays between cells. Could this be improved in the model to more closely mimic the experimental situation where delays between cells are on the order of magnitude that is the same as the order of magnitude of burst/fast oscillation durations? If there are objective reasons that this cannot be done easily with existing models (by for instance destabilizing the model with increasing heterogeneity), I suggest that the authors point out this difference between experiments and model and more explicitly address the nature of this discrepancy.

As we also responded to reviewer 3: It was not necessarily surprising to us that the computational model required a high threshold to get the power-law type distribution. Correlation coefficients are strongly decreased with stochastic noise. To illustrate this, in Author response image 1, we plot the correlation coefficient between two off phase sine waves. The first has no noise and a correlation of 0.997 and the second has 10% added noise and a correlation of 0.82. There is no noise in the computational model (because it is simulated data) while there is noise in experimental imaging data. We anticipate that this noise difference is the reason for the difference in correlation threshold. We now expand our comment on this: See Discussion page 14 lines 448-455.

**Author response image 1. sa2fig1:** 

– From the inset in Figure 1d, it is also not clearly visible what the temporal delay between traces in different cells was. Please, provide a more detailed inset/zoom-in.

Thank you for the suggestion, we have changed the figure accordingly. See Figure 1d (and also Figure 2c).

– In addition to the mean parameter values, I recommend that the authors also provide the range of values more precisely quantify heterogeneity. Since the statistics in Figure 1 are based on a rather small number of simulated islets (N=5), I suggest that effect sizes be reported as well.

We have updated the methods to provide the exact distributions used for imposing heterogeneity. The number of gap junction connections was given by a normal distribution mean 5.25, standard deviation 1.6, with the maximum of 12 and a minimum of 1 gap junction connections per cell. Cellular heterogeneity was introduced by assigning randomized metabolic and electrical parameter values to each cell based on their distributions previously determined by experimental studies. For example, coupling conductance (*g_coup_*) was assigned using a γ distribution with *k* = 4, θ = 4 and then shifted so the islet average was *g_coup_*=0.12nS. Glucokinase (*k_glyc_*), the rate-limiting step in glycolysis, was assigned using a normal distribution with mean 1.26x10^-4^ s^-1^ and standard deviation 3.15x10^-5^ s^-1^. Conductance of the KATP channel (*g_KATP_*) was given by a normal distribution with mean 2.41 pA mV^-1^ and standard deviation 0.57 pA mV^-1^. These details are all now included: See Methods page 19 lines 594-603 and 623-632.

Effect sizes are now included – See Figure 1 caption (page 34) and figure 2 caption (page 35), and Methods page 22 lines 719-720-753. This further supports the dominant association of *k_glyc_* and marginal association of *g_coup_* in hub cells in the Cha-Noma Model:

– In the model, only 500 seconds were simulated, and then the so-called fast calcium oscillations present during the second (plateau) phase of the calcium response (and brought about by bursts of electrical activity) were analyzed. In the simulated traces, they seem to be around 15-25 seconds long (Figure 1d). From the Methods section, it is not entirely clear what period of calcium traces obtained by calcium imaging in isolated islets was used for the network analysis. The authors state that "[Ca^2+^] time courses were analyzed during the second-phase [Ca^2+^] response when the slow calcium wave was established", but Figure 2 and other parts of the manuscript do not provide enough information to be sure whether the interval used for network analyses included the whole traces beyond approximately 500 seconds (Figure 2b) or only smaller parts of these traces that would correspond in duration to intervals used on simulated traces. I think it is critical that the authors address and resolve this question in detail for the following reasons.

Thank you for your careful reading and pointing this out. We have now updated the methods with the following information. See Methods page 16 lines 534-537.

“For fast oscillations, 400-500 seconds of [Ca^2+^] response was analyzed, matching the simulation time. For slow oscillations, 741-1000 seconds of [Ca^2+^] response was analyzed. This time period was chosen so at least 3 oscillations were completed. In mixed oscillations, 858 – 1000 s of [Ca^2+^] response was analyzed.”

– First, simulated traces reflect only fast oscillations, whereas in experimental traces (See Figure 2b), there are a few slow oscillations that last approximately 300 seconds each, and superimposed on them fast oscillations (approximately 10 seconds long), corresponding to oscillations analyzed on simulated traces. Since there may be fundamental mechanistic differences between fast and slow oscillations, the first being more importantly determined by ion channel properties and electrical coupling and the second more importantly by metabolism, the networks constructed from experimental traces may contain and provide information that is different from the information provided by simulated traces and networks based on them. If the authors included in their analyses on experimental traces only one part of a slow wave (e.g. 200-300 seconds) containing a few fast oscillations (e.g. 10 fast oscillations), then the simulated traces and networks can be compared with experimental traces and networks. If not, i.e., if they included longer periods (e.g. 1000 seconds) containing a few slow oscillations and thus many more fast oscillations (e.g., 50 fast oscillations), then the Pearson correlation between traces may heavily depend on the slow trends defined by slow oscillations (as addressed recently in Zmazek et al. Front Physiol 2021 and Stozer et al. Front Endocrinol 2022), and thus the network metrics convey entangled information on both fast and slow properties. Most importantly, the main conclusion from the experimental part that the rate of metabolism may be more important than other factors in determining functional network properties may be due to the prevailing influence of metabolic oscillations on networks based on experimental recordings. In this case, I suggest two possible solutions. The authors could extend the simulations to produce a mixed pattern of oscillations (which should be possible, given the existing work by Bertram, Sherman, Satin, Nunemaker, and Benninger) and then systematically analyze the impact of the fast and slow components on network properties and the importance of different parameters (kglyc, gKATP, gcoup) and compare the findings with experimental data where they could also separately study the two components. Alternatively, the authors could extract the fast component from the experimental traces (by using an appropriate filter) and limit their analyses and discussion to the fast oscillations only.

The reviewer raises an important point and caveat associated with our comparison of simulated and experimental data. This point was also raised by reviewer 4. The simulated data (Figure 1) has Ca^2+^ oscillations of ~20s period (i.e. fast oscillations) whereas the experimental data (Figure 2) has Ca^2+^ oscillations of ~300s period (i.e. slow oscillations). To address this comment, we have performed several additional experiments and analyses:

1. We collected additional Ca^2+^ (to identify the functional network and hubs) and FRAP data (to assess gap junction permeability) in islets which show either pure slow, pure fast, or mixed oscillations. We generated networks based on each time scale to compare with FRAP gap junction permeability data. We found that the conclusions of our first draft to be consistent across all oscillation types. There was no relationship between gap junction conductance, as approximated using FRAP, and normalized degree for slow (Figure 3j), fast (Figure 3 Supp 1d,e), or mixed (Figure 3 Supp 1g,h) oscillations. We also include discussion of these conclusions – See Results page 7 lines 184-186 and lines 188-191, Discussion page 12 lines 357-360.

2. We also performed additional simulations with a coupled ‘Integrated Oscillator Model’ which shows slow oscillations because of metabolic oscillations (Figure 2). We compared connectivity with gap junction coupling and underlying cell parameters. In this case, there is an association between functional and structural networks, with highly-connected hub cells showing higher gap junction conductance (Figure 2f) but also low KATP channel conductance (*g_KATP_*) (Figure 2e). However, there are some caveats to these findings – given the nature of the IOM model, we were limited to simulating smaller islets (260 cells) and less heterogeneity in the calcium traces was observed. Additional analysis suggests the greater association between functional and structural networks in this model was a result of the smaller islets, and the association was also dependent on threshold (unlike in the Cha-Noma fast oscillator model) robust. These limitations and results are discussed further (Discussion page 11 lines 344-354).

Additionally, in the IOM, the underlying cell dynamics of highly-connected hub cells are differentiated by KATP channel conductance (*g_KATP_*), which is different than in the fast oscillator model (differentiated by metabolism, *k_glyc_*). However this difference between models can be linked to differences in the way duty cycle is influenced by *g_KATP_* and *k_glyc_* (Figure 1h, Figure 2g). In each model there was a similar association between duty cycle and highly-connected hub cells. We also discuss these findings (Discussion page 11 lines 334-343).

Overall these results and discussion with respect to the coupled IOM oscillator model can be found in Figure 2, Results page 6 lines 128-156 and Discussion page 11 lines 332-354.

– Second, resolving the above point is not only important for this manuscript, but also for resolving the (apparent) differences between studies (and the Rutter-Rorsman dispute).

This is an important comment. We also hope that our manuscript will help resolve the apparent differences between studies. We were sure to cite the Rutter-Rorsman dispute in the Introduction page 3 line 63-65.

– Finally, resolving the first above point is also particularly important for extending the findings to Cx36-/- mice where fast and slow oscillations may be affected to a different extent by the absence of gap junctions, with desynchronized fast oscillations and possibly less desynchronized slow oscillations.

This is an excellent point. While homozygous Cx36ko islets lack any synchronization, a partial decrease in Cx36 such as in a heterozygous Cx36ko islet may influence fast and slow oscillations differently. Our findings show the association between β cell hubs or the functional network with gap junction coupling structural network is not dependent on oscillation timescale (Figure 3 and Figure 3 Supp 1). Thus, both fast and slow oscillating islets should show similar changes with decreased gap junction coupling. We now mention this – Discussion page 13 lines 438-440.

Reviewer #3 (Recommendations for the authors):Briggs et al. present a strong set of both computational and experimental approaches to investigate some current controversies in the functional relevance of β cell heterogeneity and pancreatic islet function. They conclude that β cell connectivity is driven by metabolic rather than gap-function-mediated structural coupling. This lays the ground for future studies understanding how metabolic coupling relates to the identity of "hub cells" and to what extent this can be targeted in the treatment of diabetes. Overall this is a very strong paper that is of interest to the readership of a journal like eLife. It is extremely well written. I would suggest that more experimental data for the NADPH and FRAP experiments might build up confidence in the finding that connectivity is defined by metabolic coupling – at present, the datasets are convincing that metabolic coupling is absent in cells that lack calcium connectivity but are not convincing enough in the opposite direction.

We thank the reviewer for their high enthusiasm for our manuscript. As detailed below we have collected and present more experimental data and also include more robust statistical analysis of the experimental data.

Here are my general commentsThe authors state that "Functional networks represent the emergent system behavior". Diabetes ensues from the autoimmune destruction of β cells (T1) or the functional demise (a mix of environmental insult with a genetic predisposition – T2). To what extent can network theory really tell us anything about the pathophysiology of these disease states or aid the development of new treatments? Or is this simply another tool, like GSIS, for assessing islet dysfunction? I think it would be useful to have this sort of oversight mentioned in the discussion, to persuade readers of the real relevance of your work.

While the islet network does provide a tool to assess overall islet function, we would suggest it also provides a means to examine individual cell sub-populations and provide insight into how they are driving islet function. We now mention in the discussion the motivation to use network theory in the context of diabetes (below). See Discussion page 10 lines 236-330.

“By understanding the formation of islet functional networks we can understand how β-cell subpopulations drive islet function. Knowledge about the role of β-cell subpopulations will also be useful when developing new treatments, such as identifying and preserving functional sub-populations in diabetes and generating stem cell derived insulin secreting cells for restoring insulin secretion.”

"cell hubs can exert a strong influence over islet dynamics21,24,25 53 and are preferentially disrupted in diabetes23" Did the Johnston paper really conclude that diabetes is causally related to the preferential loss of hubs over followers? Or is it more likely that in a dysregulated islet loss of "hubs" becomes more apparent? I personally would have prefaced this paper by summarising the mathematical and experimental evidence for hubs that exhibit small-world properties.

Thank you for pointing out a very important miscommunication in our introduction. We agree that Johnston did not show any causality and instead showed that diabetic cytokines influenced the existence of β cell hubs but did not show a preferential loss of hubs over followers. We have updated our introduction accordingly. We have also changed our introduction as you recommend to summarize the evidence more clearly for and against small-world properties. See Introduction page 3 lines 57-65*.*

When you say "structural location" what do you mean – you didn't for example look at a position relative to blood vessels, α cells, nerve endings, and δ cell projections. As a general comment, I'm not really comfortable with you equating conductance (ie the physical number of gap junction interfaces between β cells) as synonymous with their structural topography. There are so many unknowns here still – for example the contribution of other endocrine cells, nerves, and "humoral factors". Your computational model essentially randomly assigns heterogeneity in terms of metabolic rates, for example, across the 1000 "β" cells. In fact, this may not reflect the fact that such heterogeneity is indeed based on a cell's position relative to other aspects of the islet microenvironment (not just other β cells).

We have updated our manuscript to be clearer about this topic and do not use the words location or position for cells within the structural network. For example “are highly correlated subpopulations (β-cell hubs) within the β-cell functional network differentiated with respect to the structural network”. Thank you for the suggestion. See for example Introduction page 3 lines 76-78; Results page 5 lines 99-100 and throughout the manuscript.

Results section 1Briggs et al. start with data extracted from a model. Whilst the thresholds selected are generally well justified and the testing of various thresholds adds robustness, it is, in the end, just a model. It has been well validated by electrophysiologists, but the audience of this paper is wider and I think it is worthwhile reminding us in a couple of sentences of what the major inputs are into this model and how that results in oscillatory behaviour (I know you do this in supplemental anyway). Presumably, this is being run for a certain ambient glucose level ie parameters change between low and high glucose. It's important that you point out there is no accounting for other endocrine cells etc – which I think you do in the Discussion.

We have included more detailed information about the model in the results – See Results page 5 lines 92-96. We similarly describe the coupled IOM model – See Results page 6 lines 131-138.

Thank you also for pointing out the important limitations of the model that we use – and models in general. We agree and describe these limitations in greater depth within the discussion, in the “Potential Limitations” section – see Discussion page 14 lines 443-446.

Was such a high R-value necessary to get the requisite power-law type distribution a surprise to you?

As we also responded to reviewer 2: It was not necessarily surprising to us that the computational model required a high threshold to get the power-law type distribution. Correlation coefficients are strongly decreased with stochastic noise. In Author response image 1, we plot the correlation coefficient between two off phase sine waves. The first has no noise and a correlation of 0.997 and the second has 10% added noise and a correlation of 0.82. There is no noise in the computational model (because it is simulated data) while there is noise in imaging data. We hypothesize that this noise difference is the reason for the difference in correlation threshold. We now comment on this See Discussion page 14 lines 448-455

Were all 1000 cells included in your readouts?

Yes, we used 1000 cells in all five model runs. We mention this now – see Methods page 18 line 607-608.

Why did you choose the 60% connection cut-off to define hubs? Is this done by others?

The threshold was set to 60% of the max degree to align with Johnston et al.[4], which was the first paper to define hubs. We describe this now in the Methods and in the Results where we first describe the threshold cutoff. See Methods page 20 lines 661-662 and Results page 5 lines 106-108.

In your supplementary data as you move (in very tiny increments) across R thresholds the number of connections on your axis changes apparently randomly (the scale goes from 0-200 to 0-800). Why?

This is a result of the functional islet network having a roughly scale-free distribution. In Figure 1 Supplemental 1, we see that as the threshold decreases, the number of cells that are considered “high degree” will increase in a power-law like manner.

The differences in Kglyc between hubs and followers are statistically significant but the difference in absolute terms seems very small – is this relevant?

In line with the question from reviewer 2 (question 4), we outlined the distributions of the parameter values in the methods to provide more clarify about the spread of the parameter distribution. The average *k_glyc_* for non-hubs was the 1.26x10^-4^ s^-1^ (which is the average of the distribution because most cells are non hubs) while the average *k_glyc_* for hubs was 1.4x10^-4^ s^-1^ which is about half of a standard deviation higher. This is further supported all effect sizes being much greater than 0.8 for all significantly different findings (Cha Noma – *k_glyc_*: 2.85, *g_coup_*: 0.82) (IOM: *g_KATP_*: 1.27, *g_coup_*: 2.94) – We have included these effect sizes in the captions see Figure 1 and 2 captions (pages 34, 35) and Methods page 22 lines 722-723.

Results section 2This is the experimental dataset which is very interesting."We extracted the functional network (Figure 2c) and again generated a normalized degree distribution which reflects a scale free-like distribution."– ok but there is no mention of actual R values here – presumably much less than the modelled ones previously and the cut off for defining hubs on normalised degree of "edginess" is also different from the above (and others?).

We have updated the sentence to contain the threshold. We feel the network degree vs recovery plot was informative. However, we now also include a comparison between hubs vs non-hubs, where hubs are defined in the same as the computational method and Johnston et al. 2016 (e.g. nodes with >60% of maximal edges in the network). See Results page 6 line 163, page 7 line 175-176 and Figure 3d.

Conversely, you then introduce the concept of a "low degree" cell without defining that.

Following above, we include both network degree vs recovery/NADH and hub vs non-hub comparison. We now do not include any ‘low degree’ comparison as we agree this may cause some confusion.

Just out of interest, the cross-section shows that some β cells are much more fluorescent than others, presumably reflecting variation in GcAMP expression. Do you think this is an issue for your calcium trace analysis?

This is a very insightful comment. We normally see variations in the GCamP6 signal likely due to variations in GCamP6 expression (as also seen in other fluorescent-protein expressing models). However, since the time-courses are normalized to the mean GCamP6 signal we do not anticipate that this has an impact on subsequent analysis. We did analyze the relationship between GCamP6 signal and normalized degree and found no correlation (Author response image 2), which we now mention. See Results page 6 lines 166-167.

In the Methods section please can you explain in a bit more detail how you extracted your NADPH data – over what time period/resolution etc.

NAD(P)H was measured before and after the Ca^2+^ time course where glucose starts at 2mM and is elevated to 11mM. Thus the 2mM measurement is prior to Ca^2+^ imaging and the 11mM measurement after the conclusion of Ca^2+^ imaging. We also updated the methods with these details and exactly how the NAD(P)H response metric was determined – see Methods page 16 lines 517-524.

"Furthermore, the NAD(P)H response trended lower in cells that were functionally disconnected (Ca^2+^ 128 oscillations lacking any synchronization), compared to connected cells (Figure 2g)."Presumably, you only looked at calcium oscillating β cells as inactive ones will obviously likely have no other How did you define an oscillation? Your time course of over 30 minutes looks long. The association between low NADPH signal and low connectivity seems much more robust than that between hubs and high NADPH signal. Would repeat experiments firm this up?

We did only examine β cells in which GCamP6 signal was above a background threshold. However, any β-cell with a GCamP6 signal above background will be included in the analysis, irrespective of whether oscillations occurred or not. Inactive cells will likely show little coordination and thus have a low to zero network degree. Thus, we do not need to define an oscillation in our analysis.

We collect Ca^2+^ data for ~30 min in order to include sufficient numbers of oscillations to generate the functional network – this is because the oscillations are often slow (although for fast osculation we use a narrower time window). The time window used is now described – see Methods page 17 lines 537-540

We do appreciate that low connectivity does indeed appear to show lower NAD(P)H as compared to high connectivity showing higher NAD(P)H. One speculation for this is that given the power-law-like degree distribution, there are more cells with low degree than high degree in each islet. Since each dot represents the average over each islet, the lower degrees are more robust to outliers than the higher degrees. However as described above we now compare either NADH vs network degree or NADH between hub and non-hub cells. In each case we provide more robust statistical analysis and linear regression to show that a relationship between functional network degree and metabolism exists (or is trends toward signficance): for NADH hub vs non-hub (p=0.0061), for NADH vs network degree linear regression (p=0.079). See figure 3 caption (page 36) and Results page 7 lines 184-191.

Is the FRAP experiment powered? I don't have a feel for the sensitivity of this method to pick apart quantifiable differences in gap junction connections but the numbers here seem low – only 4 islets.

We have performed additional Ca^2+^ and FRAP measurements, and include data from an additional 11 islets. These data are further separated into slow, fast, and mixed oscillations and compared as either linear regression vs network degree or between hub and non-hubs (none of which show a significant relationship between network degree and gap junction permeability). See Figure 3 i,j and Figure 3 Supp 3.

Results section 3The next section poses the question what does the islet functional network indicate about its underlying structure or intrinsic properties on an individual cell basis?The authors appear to have returned to their simulated data here which initially confused me so should be headlined at the outset.

We have adjusted the Results section titles and writing to make this clearer. See Results page 7 lines 196, 230.

Given that the EPists know that GJ coupling cannot explain connectivity across more than a few cells, I think it's important to state in the main text that you enforced spatial limits on your structural connectivity analyses.

This is an important fact and we now mention this in the section “Elevated glucose metabolism is a greater driver of longer-range functional connections”. See Results 8 lines 231-233.

“In accordance with the islet cytoarchitecture, gap junction connections only exist between highly proximal cells. However, there is no distance constraint on [Ca^2+^] oscillation synchronization (Figure 5a). To determine whether this spatial constraint was responsible for the low correspondence between the functional and structural networks…”

Some of the surprising findings e.g."The probability that two cells were synchronized in the functional network, given that they shared a gap junction in the structural network, was = 0.39" has been discussed well later (eg where they don't tally with prior experimental measures.

Thank you for your comment, we elaborate on these findings in the Discussion (page 13 lines 400-410).

On the whole, I find the ending statement "These results further indicate that metabolic activity, not gap junction connections, is a greater driving factor for cells to show high [Ca^2+^] synchronization and thus influence the functional network." to be robust.

Thank you for this comment.

The next section looks at long-range functional connections which traverse cells, quite dense. I thought the experiment that modelled GJs of higher conductance to be rather extraneous and could have gone into supplemental but I don't feel strongly about this.

We have edited the section concerning long range connections so that it is more assessable. Results examining long range connections are restricted to one figure, with additional data in Figures 5 Supp. 1. However, we do feel some of these findings are important to retain as it is quite important as it removes the spatial component of the structural network.

We have now removed the figure that modelled gap junction conductance correlating with metabolism.

I think there needs to be some unpacking of the term kglyc as a measure of "glucose metabolism" – how does the sameness of kglyc translate into closer coordination of calcium oscillations? Is this simply the speed with which glucose sensing and insulin release are being cycled? Is metabolic "coupling" simply a coincidence – is this all simply an epiphenomenon?

We have more carefully defined *k*_*glyc*_ when it is first introduced in the results related to figure 1 (as also with *g*_*KATP*_) – see Results page 5 lines 110-112.

“Glucokinase conversion of glucose to glucose 6-phosphaste is the rate limiting step in glycolysis, therefore the rate of glucokinase activity (*k_glyc_*) is a proxy for the metabolic activity in the cell.”

We also clarify within the results and discussion our use of metabolism. We avoid the term “metabolic coupling” but rather describe *metabolism* as the intrinsic metabolic activity of the cell. We would in fact argue that no evidence for metabolic coupling in the islet has been presented and is not likely given the cationic selectivity of Cx36 gap junctions. See Discussion page 13 lines 395-397.

Finally, the authors look at experimental data from connexin KOs (het and homs vs WTs). I found this data (even though we have seen this before) to be rather surprising given the previous narrative. In essence, what it shows is that a 50% reduction in GJs has a marked effect, and a total loss results in a near-complete loss of hubs. If connections, and therefore presumably networks that deliver hubs, can exist just based on random metabolic homogeneity in distant β cells alone, why does this happen? Is there any NADPH data from the connexin KO islets? With the loss of gap junctions, I can see why the amplitude of the calcium oscillations in β cells might become smaller, but why do other metrics of the oscillations eg frequency seem to change?

It is well accepted that gap junctions are essential for synchronization over the entire islet and that decrease in gap junction changes the total islet behavior. We have rewritten much of the paper to clarify that gap junctions are still essential for overall islet synchronization, but they are not the primary factor driving the variations in synchronization that determines the functional network. For example, while β-cell hubs do not have higher than average gap junction conductance, they still require gap junctions to mediate the effect of their greater metabolic activity to synchronize with other β cells. This can also be seen in figure 5 where paths with higher gap junction conductance do start to show some overlap with long range functional connections. For example, see Results page 9, line 256.

Frequency is known to be affected by gap junctions conductance because as gap junction conductance decreases, the cells’ intrinsic frequency (rather than collective frequency from islet communication) dominates [9], [10].

FiguresFigure 1 – Kglyc and g coup need to be defined/explained so the figure stands alone.

We have adjusted the manuscript accordingly – See Figure 1 and 2 captions (page 34, 35).

Figure 1b – the word seed is confusing/comes out of the blue. Do you actually mean "modelled islet"?

This is a great point – the seed is a technical word for how the random number generator is defined in order to replicate the simulated islet experiments. However it is unnecessary to use this terminology outside of the methods. All references to “seed” are now “simulated islet” (see Figure captions)

Figure 1 f/g – this may display my incomplete understanding of the different populations you defined and measured, but why was a paired t-test used to compare them and if these readings are truly paired were they really normally distributed?

Thank you for this comment. The paired t-test is used because each comparison comes from the same simulated islet. To ensure robustness, I repeated the analysis with unpaired t-test and the interpretation did not change (see Author response table 1). I also tested for normality using Shapiro-wilk and Kolmogorov-Smirnov. All samples fit the threshold of normality except for non-hubs which tested normal under KS but not Shapiro-Wilk.

**Author response table 1. sa2table1:** 

	UnpairedP-value	Pairedp-value	Shapiro-Wilk Normality Test (Yes = normally distributed)(Hubs/Non-hubs)	Kolmogorov-Smirnov Normality Test(Yes = normally distributed)(Hubs/Non-hubs)
kglyc	<0.0001	0.0001	Yes/Yes	Yes/Yes
gcoup	0.0180	0.0147	Yes/Yes	Yes/Yes
gKATP	0.1602	0.1829	Yes/No	Yes/Yes

Figure 2 – the defined cellular ROIs in a is not the same as the ones shown on your map in c – I was expecting them to be!

Thank you so much for your careful observation. The map in c was rotated incorrectly. We have corrected the figure so that the ROIs in Figure 3a (previous figure 2) correspond to the network in Figure 3c.

Again, here we have the notion of Kmax mentioned which needs defining in the legend.

Thank you, we have removed the use of Kmax and redefined the legend as 90% of the max degree. (Figure 3 caption, page 36)

Reviewer #4 (Recommendations for the authors):1. The paper is very dense and complex, and thus is difficult to follow, and as a consequence will only be understandable to a very specific and likely small group of readers. The methods, the analysis, and the models used, while largely based on prior work require careful reading, and a true understanding of their implications will be mostly lost on most readers. The paper would therefore be improved by reducing its size and complexity and removing excessive verbiage. Some of the more mathematical aspects could be relegated to either previously published papers by the group or placed in the Supplementary Material. It might be worth considering breaking the paper into two different but complementary ones, one emphasizing theory and the other measurements but I realize this would make reading the two simultaneously very difficult.

Thank you for this suggestion. We agree that the manuscript is quite dense. We have made the following changes to make the manuscript more accessible. In consultation with the editor, we decided not to break the paper into two given journal policies; but rather focus on eliminating more peripheral data. Similarly, the mathematical parts of the methods we retain in the methods but are happy to move if the editor suggests this.

1. Significantly shortening of the Introduction and rewriting of the Discussion.

2. Substantially reducing the last two figures from the previous manuscript (figure 5,6) and reducing the associated results text- this includes removing the complex network metrics and kept only clustering coefficient and global efficiency (see Figures 6, 7).

3. Removing simulated islet data and associated Results text where metabolic activity and gap junction coupling were correlated.

2. The paper would also be improved by very clearly highlighting what is truly new in the paper and deemphasizing what is a restatement of what has been known already (e.g. discussing how the islet can be considered a small world network, etc).

Thank you for the suggestion, we have made significant modifications throughout the Introduction, Discussion and Results to be clearer about what is known from previous work and what is newly found in this manuscript.

Reviewer #5 (Recommendations for the authors):I have a major concern about how cells related by cell metabolism and cells related by gap junctional coupling are treated as independent. There is no mechanism mentioned (?) for metabolism to synchronize islet cells independent of gap junctions. Consequently, I have concerns about the conclusions of the results.

Thank you for this comment and valid point of concern apologize for the confusion. We have tried to make the mechanisms that we are referring to much clearer. Our study is not assessing metabolic coupling or any means of metabolic intermediates actually moving between cells to synchronize them. Rather, we are looking at how to interpret the functional network which shows cells that are highly synchronized. The functional network is based on how synchronized β cell pairs are. Given that β cells are intrinsic oscillators, if two cells have identical intrinsic oscillations (which is primarily driven by the metabolic activity of the cell), then the cells will readily “synchronize” and have an edge in the functional network with minimal gap junction mediated current. Thus two cells can become highly synchronized without necessarily showing high levels of gap junction coupling. This explains why in our partial gap junction knock out experiments (figure 5 and 6), some cells are still highly connected in the functional network. Our aim in this paper was to show that the functional network also reflects the intrinsic cellular characteristics rather than mainly gap junction coupling. As an extreme example, in the complete absence of gap junction coupling a few cells that show identical Ca^2+^ oscillations and will therefore show a functional edge despite lacking any gap junction connections. We have tried to make this more clear in the paper – see Discussion page 13 lines 395-397.

This work seems to suggest a disregard of the necessity of gap junctional connections for islet synchronization, and suggest that functional connectivity – based on statistical correlation of traces – somehow predominate…up until Figure 6 where knocking down gap junctions is acknowledged as important.

We apologize again for the confusion. We have tried to emphasize the important role of gap junctions in whole islet synchronization to clarify the aim of this paper, which we also attempt to clarify in the reviewer's first comment. Results (page 7 lines 203-205, page 8, lines 241-242, page 9 lines 256 and 293-294, page 10 lines 308-309) and Discussion (page 11 line 318-319, page 13, lines 434-435).

I think identifying subpopulations is reasonable, but when a 60% functional connection threshold is what defines "hub" cells and that produces between 50 and 200 cells, "hub" seems to lose its meaning,

Thank you for the necessary point of clarification. We agree that 60% of functional connections could be large depending on how a hub cell is to be interpreted. In this paper, we chose to use 60% in order to directly match methodology used in Johnston et al[4], which is the first paper to describe β cell hubs. This logic is noted in Methods page 20 lines 661-662 and Results page 5 lines 106-108.

However, given that the degree distribution is roughly logarithmic, 60% of the maximum degree produces a relatively small number of hubs per islet. As you note, the total number of hub cells in a 1000 cell islet was: 52, 47, 109, 88, and 120, which is equivalent to 5%, 4.7%, 10%, 8.8%, and 12% of the total number of cells in the islet.

especially when these "hubs" seems to be all grouped together rather than dispersed as might be expected in a small world network. The small network-ness is questionable.

Thank you for bringing up a very interesting observation – that “hub” cells tend seem to be located towards the center of the islet. We do not have a mechanistic understanding of this observation because the locality of the hubs cells was not a focus of this paper but could be looked at in subsequent work. Indeed prior experimental measurements in Johnston et al[4] suggested that there was not a spatial dependence.

However, we acknowledge the reviewers point that it is strange the cells tend to be grouped together. However, the functional network (from which the small world-ness has been suggested by Stozer et al[1]) is not spatially constrained. Therefore even if the hub cells are physically grouped, they may be more dispersed in the functional network. Additionally, it is a well-known phenomenon in scale-free networks (or roughly scale-free networks) that high degree nodes (hub cells) tend to have be highly connected with other high degree nodes. This comes from the so called “friendship paradox”[15].

While we are happy to further improve our measurement of the small word-ness (beyond the methods we previously used), in order to make the manuscript more accessible and focus on the more important findings of the manuscript, we have de-emphasized the small world-ness results.

The sensitivity to the correlation threshold suggests a certain lack of robustness. Perhaps I am misunderstanding the number of cells labeled as "hub cells."

We completely agree with the reviewer that the sensitivity to the correlation threshold indicates a lack of robustness. This sensitivity would occur in any study which uses functional network analysis, which is one of the reasons that we chose to explicitly analyze the sensitivity. However, in this manuscript, we are not attempting to show that the islet functional network is scale free or small world. These results have been demonstrated in previous papers[1], [4] and our threshold was chosen to match these claims.

Rather, this manuscript is attempting to show that cell metabolic activity is more highly associated with the functional network than is the gap junction structure. We found that in both the Cha-Noma model and experimental measurements, results concerning metabolic activity, and structure were not sensitive to threshold, E.g. hub cells were more differentiated by *k_glyc_* (glucokinase rate) than *g_coup_* (gap junction conductance) for all thresholds analyzed – See Figure 1 Supp. 1 and Figure 3 Supp 1.

The excitability of cells via metabolism, for example, must be communicated through some (often structural) mechanisms. I appreciate that this article is attempting to get at how that division breaks down and I think much of the calculation and simulation is useful, but the language expressing the certainty rather than the more appropriate equivocation is not completely appropriate.

We agree that metabolic coupling or communication of metabolites between cells must occur through some type of structural mechanism. Hopefully, our clarification from the reviewers first two questions as well as rewording in the manuscript, clarifies that we are not attempting to describe sharing of metabolism between cells but rather investigating similarities in metabolic rates. We clarify this more explicitly now – For example, see Discussion page 13 lines 395-397.

[Editors’ note: what follows is the authors’ response to the second round of review.]

The manuscript has been improved but there are some remaining issues that need to be addressed, as outlined below:Reviewer #2 (Recommendations for the authors):I much appreciate the effort that the authors put into addressing my main skepticism about the paper, namely the fact that one type of oscillations were analyzed for experimental data and another type for modelled data, by including an additional model for slow oscillations and additional experimental data.Given the new results, below, I provide my additional comments and suggestions that should not be too difficult for authors to consider and take into account.

We thank the reviewer for their positive comments regarding the revisions we made to the manuscript. We appreciate the additional comments and suggestions and have addressed them following the comments below. Please note that the *eLife* pdf conversion may result in some line and page numbers changing in comparison to the uploaded word version of the manuscript.

110-197 and Figures 1-S3:First, the difference between hubs and non-hubs in terms of kglyc is around 0.14 vs 0.13, whereas for gcoup it is around 0.75 vs 0.6. In relative terms, the former is an 8 % difference and the latter a 25 % difference. It is true that the former is more significant statistically, but this is due to the fact that the heterogeneity is larger in the distribution of gcoup for hubs! Therefore, to me, it seems that both kglyc and gcoup seem to define the hub cells to a comparable degree. This conclusion seems to also be more compatible with findings for slow oscillations where the relative difference between hubs and non-hubs for gcoup is again around 20-30 %.I suggest that authors consider this under the corresponding sections of results and discussion, and perhaps to some extent also in Figure 8.

We appreciate the reviewers point with respect to the relative differences in gcoup and kglyc between hubs and non-hubs, and that indeed the greater variability in gcoup than kglyc could perhaps masks how each parameter may define hubs. We now explicitly state the values of kglyc, gKATP, and gcoup between hubs and non-hubs in the respective Results section for figure 1 (where the differences are 11% for kglyc and 15% for gcoup) and for figure 2. However, we do note that the gcoup differences are less robust, in that they do not hold for all Rth values used. We briefly mention this higher difference for gcoup, but has greater variability and is less robust in the Discussion. Given this and that experimental results do not support a difference in gcoup but do with kglyc, we do not feel it warranted to adjust the figure 8 to include a role of gcoup. See Results page 5 line 114-117, page 6 line 146-148 and Discussion page 11 lines 360-362.

Second, at a threshold for hub cells of a 60 % normalized degree (i.e., the criterion for the separation line between hubs and non-hubs), different percentages of β cells qualify as hubs for different simulated islets, as evident in Figure 1b. More specifically, the cumulative number of cells qualifying as hubs for islets 3 and 5 is much more than the cumulative number of hubs in simulated islets 1 and 2. This differs from the criterion used in other important recent studies, i.e., in Johnston et al. Cell Metab 2016, hubs typically represent 1-10 % of islet cells, in Lei et al. Islets 2018, hubs represent 10 % of cells, and in Stozer et al. Am J Physiol 2021 and in Sterk et al. Biophys J 2023, hubs represent 1/6th of islet cells. I think it would enhance the comparison between the present and previous research if the authors provided the % of hub cells per islet.

We thank the reviewer for raising these interesting and important points. By using the same criterion as in Johnston et al. for defining hub cells, we do indeed observe the % of hub cells to vary between 5-12% for fast oscillating simulated islets (Figure 1b); to vary between 3-7% for slow oscillating simulated islets (figure 2b); and to vary between 3%-40% for experimentally measured islets (figure 3d). We now include these values in the Results section to indicate the range of hub cells observed under each condition using this criterion. We do note that in some cases (mainly experimental cases) this range is greater than the previously noted proportions (1-10% or 16%) and this is a good motivation to test whether our findings hold for an alternative definition for hub cells in the next point. See Results page 5 line 108, page 6 line 145 and page 7 line 175.

Additionally, setting the line of separation between hubs and non-hubs at the given threshold (not changing the threshold and thus the distribution of normalized degree) individually for each islet to achieve a fixed percentage of hub cells per islet close to 10 % of the most connected cells, would be a valuable addition to supplemental figure 1 and could perhaps help detect a larger (or smaller) difference between this more extreme group of cells and the majority of non-hub cells, in terms of kglyc, gkatp, and gcoup. Performing this additional analysis at least for the threshold value used in the main Figure 1 could add much value to the manuscript and make the findings even more directly comparable with the aforementioned studies. The same suggestion could also be taken into account for the analyses of modelled slow oscillations and for the experimental analyses of metabolic activity and coupling, where 10 % of the most connected cells could be considered/classified as hubs as well. Such additional analysis should be easily feasible since it considers a fraction of cells that have already been used in the analyses, but their degree of "hubness" is probably more.

We agree another valid way to look at the most connected hub-like cells would be to examine the top 10% most connected cells, especially give that in some cases the range of hubs cells is greater than previously noted proportions. We have performed such analysis for simulated islets presented in figure 1 and figure 2, and compared *k_glyc_*, *g_KATP_* and *g_coup_* parameters between hubs and non-hubs. We also performed such analysis for experimentally measured islets presented in figure 3 and compared NAD(P)H and gap junction permeability. In each case when comparing the top 10% connected cells with remaining 90% of cells we generally observed similar findings to those presented in figures 1-3. In figure 1 (coupled Cha-Noma) *k_glyc_* is significantly different, *g_coup_* is slightly different and *g_KATP_* is similar between hubs and non-hubs. In figure 2 (coupled IOM) *k_glyc_* is similar, *g_coup_* is significantly different and *g_KATP_* is similar between hubs and non-hubs. In figure 3 (experimental) NAD(P)H (metabolic activity) is significantly different between hubs and non-hubs, whereas recovery rate (gap junction coupling) is similar between hubs and non-hubs. We now incorporate these results in new supplemental figures supporting figure 1, figure 2 or figure 3. See Results page 5-6 lines 126-129, page 6 lines 150-151, page 7 lines 186-188 and 202-204; and Figure 1 supplemental 2, Figure 2 supplemental 2, Figure 3 supplemental 2.

Given that the duty cycle (percentage of active time) so strongly correlates with the role of hub cells, it is somewhat surprising that the authors do not mention in the discussion that this same finding has recently been obtained for both mouse and human islets (Šterk et al. Biophys J 2023, Stožer et al. Am J Physiol 2021, Gosak et al. Diabetes 2022).

We apologize for omitting citations to important literature that also observes a relationship between duty cycle and node degree. Our intention was for this relationship to explain how the parameters influence hubs in each model. But this literature is important to cite to provide context that it is a well observed relationship, and we now do so. See Discussion page 11 line 356-357.

Third, I do value the effort of authors in replying to my comment regarding the exceedingly high correlation between simulated traces compared with experimental traces. However, besides noise, one very important aspect is the choice of model and the parameters which determine the observed lags between different cells or the wave speed. For instance, Cappon and Pedersen in their Chaos article built on the model used in Benninger Biophys J 2008 to produce lags between cells that are very realistic compared with experimentally observed values, whereas lags in the present study are just a fraction of the duration of a burst (fast Ca oscillation).At present, I can only speculate about that, but the rather low conditional probability that two cells sharing gap-junctional conditions also show a functional connection could also be a consequence of the exceedingly short time lags between signals in the model employed in this study. More specifically, in the model by Cappon and Pedersen, there are waves travelling across islets and direct neighbors have shorter phase-lags between their signals compared with more distant cells and therefore on average more similar signals, similar to experimental recordings in isolated islets and tissue slices (the velocity being around 100 um/s, the time lag between direct neighbors is around 0.1 s and the time lag between most distant cells is around 1-2 s). This means that the values of R will be higher for direct neighbors and at any given threshold they will be functionally connected with a higher probability than with more distant cells where lags can be an order of magnitude more.Therefore, I would suggest that in the future (not in this study), authors also repeat some of the analyses with modeled traces that show larger lags. This would probably also enable them to explore the relationship between different modelled/analyzed parameters and the role of pacemakers, i.e., wave initiators and other populations of cells. Perhaps, if authors consider the above suggestion as useful, they could include it in the discussion as a possible drawback of the present model and as a suggestion for future studies.

We thank the reviewer for their careful explanation of this important point. Indeed, it will be important in future work to compare simulated parameters between wave initiators or other populations of cells. We do note we previously analyzed properties of wave initiators in simulated islets e.g. Dwulet et al. PLOS Comp. Bio. (2021), following the same coupled Cha-Noma model. However, that analysis lacked a comparison with the functional network and thus future work should indeed examine this connection more deeply upon varying the simulated islet, such as to extend the time lags between cells. As such we mention this in the Discussion as a goal for future work. See Discussion page 14 lines 486-492.

Reviewer #5 (Recommendations for the authors):The authors have thoroughly responded to my (and others') reviews. I think their modifications are reasonable and have tempered the language appropriately. The results are somewhat interesting and the techniques employed to measure and analyze the islets including adding an additional model system are extensive. They have addressed my points.

We thank the reviewer for their positive comments regarding the revisions we made to the manuscript. We appreciate the additional point they make which we have addressed below.

One point: We discussed (Reviewer #5 question/response 5b) the potential outlier at 2.35 of Figure 1f (gkatp data – Β Cell Hubs). That data point no longer exists on the graph (that I can see?). Since they argue convincingly that point should in fact not be treated as an outlier, I expect it is an oversight of it not being there. This should be fixed (or explained) prior to moving forward with publication.

The thank the reviewers for making this point. The data point in question is slightly above 2.35 and therefore as we adjusted the y-axis scale it was mistakenly placed out of the range. We have now corrected this oversight by adjusting the y-axis scale. See Figure 1f.